# BARNN: A Bayesian Autoregressive and Recurrent Neural Network

**Dario Coscia**[1,2] **Max Welling**[2] **Nicola Demo**[3] **Gianluigi Rozza**[1]

## Abstract

Autoregressive and recurrent networks have achieved remarkable progress across various fields, from weather forecasting to molecular generation and Large Language Models. Despite their strong predictive capabilities, these models lack a rigorous framework for addressing uncertainty, which is key in scientific applications such as PDE solving, molecular generation and Machine Learning Force Fields. To address this shortcoming we present BARNN: a variational Bayesian Autoregressive and Recurrent Neural Network. BARNNs aim to provide a principled way to turn any autoregressive or recurrent model into its Bayesian version. BARNN is based on the variational dropout method, allowing to apply it to large recurrent neural networks as well. We also introduce a temporal version of the "Variational Mixtures of Posteriors" prior (tVAMP-prior) to make Bayesian inference efficient and well-calibrated. Extensive experiments on PDE modelling and molecular generation demonstrate that BARNN not only achieves comparable or superior accuracy compared to existing methods, but also excels in uncertainty quantification and modelling long-range dependencies.

## 1. Introduction

Autoregressive and recurrent models have demonstrated impressive advancements in different disciplines, from weather prediction (Bodnar et al., 2024; Lam et al., 2022), to molecules generation (Shi et al., 2020; Simm et al., 2020) and Large Language Models (LLMs) (Bengio et al., 2000; Vaswani, 2017; Radford et al., 2019). However, while autoregressive models show strong predictive abilities, they are also prone to overfitting to the specific tasks they are trained

on, challenging their application outside their training domain (Papamarkou et al., 2024). Overcoming this behaviour is critical, not only in scientific applications where phenomena can show complex data distribution shifts away from the training data, but also in deep over-parametrized models. For example, in weather forecasting, rapidly changing climate patterns can cause significant deviations from the training data and lead to unpredictable results, while over-parametrized LLMs often provide incorrect answers with high confidence, highlighting issues with model calibration (Jiang et al., 2021; Xiao et al., 2022; Yang et al., 2022). This problem is also prevalent across many other domains, and Bayesian approaches present a promising direction for improvement (Papamarkou et al., 2024).

This work aims to close the gap between autoregressive/ recurrent models and Bayesian methods, presenting a fully Bayesian, scalable, calibrated and accurate autoregressive/ recurrent model, named BARNN: *Bayesian Autoregressive and Recurrent Neural Network*. BARNN provides a principled way to turn any autoregressive/ recurrent model into its Bayesian version. In BARNN the network weights are evolved jointly with the observable states (e.g. language tokens, PDE states, etc.), creating a joint probabilistic model (see Figure 1). A new state is generated by sampling from the distribution conditioned on previous states and the current network weights. At the same time, the latter are drawn from a variational posterior distribution that is conditioned on the previous states. We derive a variational lower bound for efficiently training BARNN that is related to the VAE (Kingma & Welling, 2014) (ELBO) objective. To make Bayesian Inference over a large number of network weights computationally efficient, we propose an extension of Variational Dropout (Kingma et al., 2015; Gal & Ghahramani, 2016), which also opens the door to explore quantization and network efficiency (Louizos & Welling, 2017; Van Baalen et al., 2020) for overparametrized autoregressive models. Finally, we introduce a temporal version of the "Variational Mixtures of Posteriors" (Tomczak & Welling, 2018) prior (tVAMP-prior) to make Bayesian inference efficient and well-calibrated.

Building on the recent successful application of autoregressive and recurrent models to Scientific Machine Learning tasks (Pfaff et al., 2021; Lippe et al., 2024; Segler et al., 2018) we apply our methodology to *uncertainty quantifica-*

---

[1]Mathematics Area, International School of Advanced Studies, Italy [2]Informatics Institute, University of Amsterdam, The Netherlands [3]FAST Computing Srl, Italy. Correspondence to: Dario Coscia <dario.coscia@sissa.it>.

*Proceedings of the 42$^{nd}$ International Conference on Machine Learning*, Vancouver, Canada. PMLR 267, 2025. Copyright 2025 by the author(s).

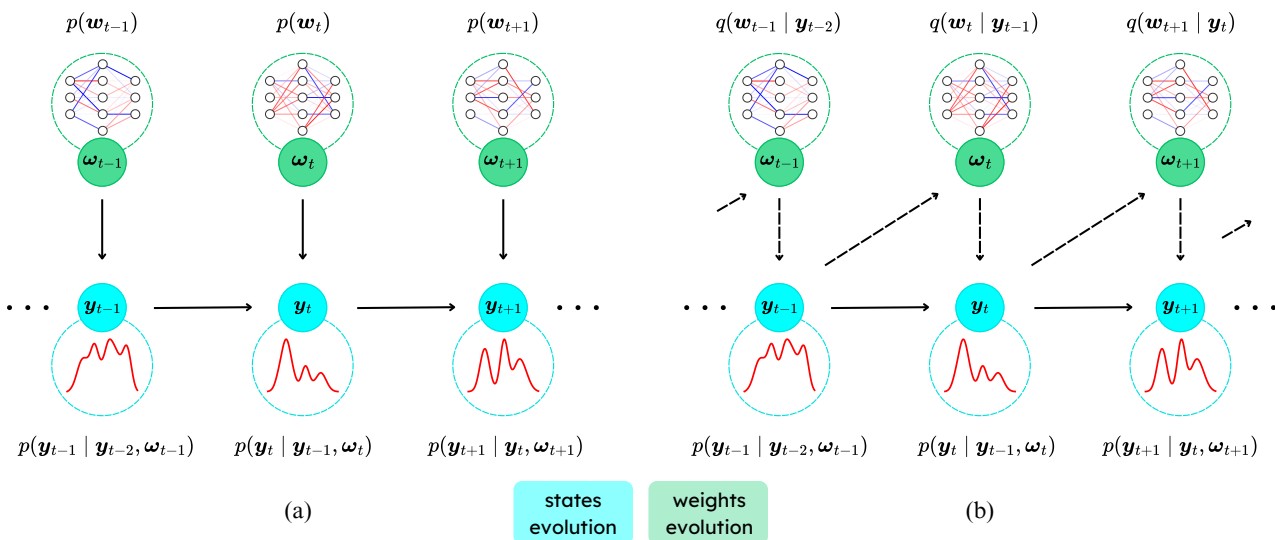

Figure 1: BARNN generative model (a) and inference model (b). Solid lines represent the generative process, whereas dotted lines indicate the variational approximation used for inference. The network weights $\boldsymbol{\omega}_t$ evolve jointly with the states $\boldsymbol{y}_t$, forming a coupled probabilistic model. During inference, the weights are sampled from the variational posterior $q$, allowing for improved parameter estimation based on observed data. The draw depicts the one-step causal evolution of the states.

*tion* (UQ) for Neural PDE Solvers, and to *molecules generation* with Language models for drug discovery. Experiments demonstrate that BARNN not only offers greater accuracy but also provides calibrated and sharp uncertainty estimates, and excels in modelling long-range molecular dependencies compared to related methods. To the best of our knowledge, BARNN is the first approach that transforms any autoregressive or recurrent model into its Bayesian version with minimal modifications, ensuring greater accuracy compared to its not Bayesian counterpart and providing a structured way to quantify uncertainties.

## 2. Background and Related Work

### 2.1. Autoregressive and Recurrent Networks

Autoregressive and Recurrent models represent a joint probability distribution as a product of conditional distributions, leveraging the probability product rule (Bengio et al., 2000; Uria et al., 2016). In recent years, autoregressive and recurrent models have been successfully applied in various to different tasks, such as text (Bengio et al., 2000; Vaswani, 2017), graphs (Li et al., 2018; Liu et al., 2018), audio (Dieleman et al., 2016), and, more recently, PDE modelling (Brandstetter et al., 2022a;b) and molecules generation (Segler et al., 2018; Özçelik et al., 2024; Schmidinger et al., 2024). Despite their vast applicability, those models are known to overfit, leading to unreliable predictions, especially in over-parameterized regimes, such as modern LLMs, or when the testing data distribution shifts signifi-

cantly from the training data distribution (Papamarkou et al., 2024). However, so far in the literature, it appears that quantifying the uncertainty of these model parameters, namely *epistemic uncertainty*, is far from being the norm, and principled ways to do it are still missing (Aichberger et al., 2024).

### 2.2. Bayesian Modelling and Uncertainty Quantification

Bayes' theorem (Bayes, 1763) offers a systematic approach for updating beliefs based on new evidence, profoundly influencing a wide array of scientific disciplines. In Deep Learning (DL), Bayesian methods have extensively been applied (Hinton & Van Camp, 1993; Korattikara et al., 2015; Graves, 2011; Kingma & Welling, 2014). They provide a probabilistic treatment of the network parameters, enable the use of domain knowledge through priors and overcome hyper-parameter tuning issues through the use of hyper-priors (Papamarkou et al., 2024). Bayesian Models have been historically applied to model epistemic uncertainty, i.e. uncertainty in network parameters. For example in (Gal & Ghahramani, 2016) the Dropout (Srivastava et al., 2014) method was linked to Gaussian Processes (Rasmussen & Williams, 2005) showing how to obtain uncertainties in neural networks; while in (Kingma et al., 2015) the authors connect Gaussian Dropout objectives to Stochastic Gradient Variational Bayesian Inference, allowing to learn dropout coefficients and showing better uncertainty. This last method is the most related work to ours, but differently from our approach, the weights do not evolve in time, leading to less

accurate uncertainties (see the experiment section 4.2).

## 2.3. Neural PDE Solvers

A recent fast-growing field of research is that of surrogate modelling, where DL models, called *Neural PDE Solvers*, are used to learn solutions to complex physical phenomena (Li et al., 2020; Bhattacharya et al., 2021; Rozza et al., 2022; Pichi et al., 2024; Coscia et al., 2024; Li et al., 2024). In particular, Autoregressive Neural PDE Solvers (Brandstetter et al., 2022b; Sanchez-Gonzalez et al., 2020) gained significant attention since they can be used to infer solutions of temporal PDEs orders of magnitude faster compared to standard PDE solvers, and generate longer stable solutions compared to standard (not-autoregressive) Neural Operators. However, Autoregressive Neural PDE Solvers accumulate errors in each autoregressive step (Brandstetter et al., 2022b), slightly shifting the data distribution over long rollouts. This yields inaccurate solutions for very long rollouts, and methods to quantify the uncertainty have been developed: PDE Refiner (Lippe et al., 2024) obtains the PDE solution by refining a latent variable starting from unstructured random Gaussian noise, similarly to denoising diffusion models (Ho et al., 2020); while in GraphCast (Lam et al., 2022) the authors introduced a method called Input Perturbation, which adds small random Gaussian noise to the initial condition, and unroll the autoregressive model on several samples to obtain uncertainties.

## 2.4. RNN for Molecule Generation

Drug design involves discovering new molecules that can effectively bind to specific biomolecular targets. In recent years, generative DL has emerged as a powerful tool for molecular generation (Shi et al., 2020; Eijkelboom et al., 2024). A particularly promising approach is the use of Chemical Language Models (CLMs), where molecules are represented as molecular strings. In this area, RNNs have shown significant potential (Segler et al., 2018; Özçelik et al., 2024; Schmidinger et al., 2024). In the experiments section, we will demonstrate how classical RNNs, when combined with BARNN, can enhance performance—particularly by improving the model's ability to capture long-range dependencies, generating statistically more robust molecules, and achieving better learning of molecular properties

## 3. Methods

Bayesian Autoregressive and Recurrent Neural Network (BARNN) is a practical and easy way to turn any autoregressive or recurrent model into its Bayesian version. The BARNN framework creates a joint distribution over the observable states (e.g. language tokens, PDE states, etc.) and model weights, with both alternatingly evolving in time

(section 3.1). We show how to optimize the model by deriving a novel variational lower bound objective that strongly connects to the VAEs (Kingma & Welling, 2014) framework (section 3.2). Finally, we present a scalable variational posterior and prior for efficient and scalable weight-parameter sampling (section 3.3).

## 3.1. The State-Weight Model

Autoregressive models represent a joint probability distribution as a product of factorized distributions over *states*:

$$p(\boldsymbol{y}_0, \ldots, \boldsymbol{y}_T) = \prod_{t=1}^{T} p(\boldsymbol{y}_t \mid \boldsymbol{y}_{t-1}, \ldots, \boldsymbol{y}_0), \qquad (1)$$

where a new state $\boldsymbol{y}_t$ is sampled from a distribution conditioned on all the previous states $\boldsymbol{y}_{t-1}, \ldots, \boldsymbol{y}_0$, and $T$ is the state-trajectory length. In particular, deep autoregressive models (Schmidhuber et al., 1997) approximate the factorized distribution $p(\boldsymbol{y}_t \mid \boldsymbol{y}_{t-1}, \ldots, \boldsymbol{y}_0)$ with a neural network with optimizable deterministic weights $\boldsymbol{w}$. We construct a straightforward Bayesian extension of this framework by jointly modelling states $\boldsymbol{y}_t$ and weights $\boldsymbol{\omega}_t$ via a *joint* distribution $p(\boldsymbol{y}_0, \boldsymbol{\omega}_1, \boldsymbol{y}_1, \boldsymbol{\omega}_2, \boldsymbol{y}_2, \ldots, \boldsymbol{\omega}_T, \boldsymbol{y}_T) = p(\boldsymbol{y}_{0:T}, \boldsymbol{\omega}_{1:T})$ on states and weights, accounting for the variability in time for both[1]. The full joint distribution is given by:

$$p(\boldsymbol{y}_{0:T}, \boldsymbol{\omega}_{1:T}) = \prod_{t=1}^{T} p(\boldsymbol{y}_t \mid \boldsymbol{y}_{t-1}, \ldots, \boldsymbol{y}_0, \boldsymbol{\omega}_t) p(\boldsymbol{\omega}_t). \quad (2)$$

Hence, a new state $\boldsymbol{y}_t$ is obtained by sampling from the distribution conditioned on previous states and the current weights $\boldsymbol{\omega}_t$, while the latter are sampled from a prior distribution $p(\boldsymbol{\omega}_t)$, see Figure 1.

## 3.2. The Temporal Variational Lower Bound

Learning the model in eq. (2) requires maximising the log-likelihood $\log p(\boldsymbol{y}_{\geq 0})$. Unfortunately, directly optimizing $\log p(\boldsymbol{y}_{0:T})$ by integrating eq. (2) over the weights, or by expectation maximization is intractable; thus we propose to optimize $\log p(\boldsymbol{y}_{0:T})$ by variational inference (Kingma & Welling, 2014). The variational posterior over network weights given the states $q_{\boldsymbol{\phi}}(\boldsymbol{\omega}_{1:T} \mid \boldsymbol{y}_{0:T})$ is parametrized by (different time-independent) weights $\boldsymbol{\phi}$ and reads:

$$q_{\boldsymbol{\phi}}(\boldsymbol{\omega}_{1:T} \mid \boldsymbol{y}_{0:T}) = \prod_{t=1}^{T} q_{\boldsymbol{\phi}}(\boldsymbol{\omega}_t \mid \boldsymbol{y}_{t-1}, \ldots, \boldsymbol{y}_0). \quad (3)$$

Given the variational posterior above, in Appendix A.1 we derive the following variational lower bound to be maximised over the variational parameters:

$$\mathcal{L}(\boldsymbol{\phi}) = \mathbb{E}_{t \sim \mathcal{U}[1,T]}[\mathbb{E}_{\boldsymbol{\omega}_t \sim q_{\boldsymbol{\phi}}}[\log p(\boldsymbol{y}_t \mid \boldsymbol{y}_{0:t-1}, \boldsymbol{\omega}_t)] \\ - D_{KL}[q_{\boldsymbol{\phi}}(\boldsymbol{\omega}_t \mid \boldsymbol{y}_{0:t-1}) \| p(\boldsymbol{\omega}_t)]]. \quad (4)$$

---

[1]We indicate a sequence $(\boldsymbol{a}_k, \boldsymbol{a}_{k+1}, \boldsymbol{a}_{k+2}, \ldots, \boldsymbol{a}_{k+l})$ with $\boldsymbol{a}_{k:k+l} \; \forall l > k \in \mathbb{N}$.

The equation above has two nice properties. First, it resembles the VAE objective (Kingma & Welling, 2014) creating a connection between Bayesian networks and latent variable models. Second, BARNN incorporates into the ELBO the timestep dependency, allowing for adjustable weights in time which we will show provide sharper, better calibrated uncertainties as well the ability to better capture long range dependencies in the sequences. Finally, in the limit of time-independent peaked distribution with prior and posterior perfectly matching, the BARNN loss simplifies to the standard log-likelihood optimization used in standard autoregressive and recurrent models (see Appendix A.5 for derivation). This suggests interpreting eq. (4) as a form of Bayesian weight regularization during autoregressive model training. Finally, once the model is trained, the predictive distribution and uncertainty estimates can be easily computed by Monte Carlo sampling (see Appendix A.4).

### 3.3. Variational Dropout Approximation

When examining the generative model in eq. (2) or the lower bound in eq. (4), a key challenge arises: how can we efficiently sample network weights? Directly sampling network weights is impractical for large networks, making alternative approaches necessary. Variational Dropout (VD) (Kingma et al., 2015; Molchanov et al., 2017) offers a solution by reinterpreting traditional Dropout (Srivastava et al., 2014)—which randomly zeros out network weights during training—as a form of Bayesian regularization, and use the *local reparametrization trick* for sampling only the activations resulting in computational efficiency. We re-interpret VD for sampling dynamic weights $\boldsymbol{\omega}_t$ during training and inference. Our goal is to do a reparametrization of the weights $\boldsymbol{\omega}_t = f(\boldsymbol{\Omega}, \boldsymbol{\alpha}_t, \boldsymbol{\epsilon})$, with $\boldsymbol{\Omega}$ static-deterministic network weights, $\boldsymbol{\alpha}_t$ deterministic time-dependent variational weights, $\boldsymbol{\epsilon} \sim p(\boldsymbol{\epsilon})$ some random noise, and $f$ a differentiable function. This parametrization is very flexible: (i) it allows to apply the global-reparametrization trick (Kingma & Welling, 2014); (ii) it uses fixed network weights $\boldsymbol{\Omega}$ enabling to scale the Bayesian methodology to large networks; (iii) it incorporates time and previous state dependency by *perturbing* $\boldsymbol{\Omega}$ with deterministic time-dependent variational weights $\boldsymbol{\alpha}_t$, and adds stochasticity with random noise $\boldsymbol{\epsilon}$, making the methodology easily implementable with little change to the main autoregressive or recurrent model. We present a specific method for defining this reparameterization while leaving other potential approaches and extensions for future work.

**Variational Posterior:** First, assume that weights $\boldsymbol{\omega}_t$ factorize over layers $l \in (1, \ldots, L)$ for the posterior $q_\phi$, implying:

$$q_\phi(\boldsymbol{\omega}_t \mid \boldsymbol{y}_{0:t-1}) = \prod_{l=1}^{L} q_\phi(\boldsymbol{\omega}_t^l \mid \boldsymbol{y}_{0:t-1}), \qquad (5)$$

where $\boldsymbol{\omega}_t^l$ are the weights at time $t$ for layer $l$ of the autoregressive or recurrent network. Second, assume $q_\phi(\boldsymbol{\omega}_t^l \mid \boldsymbol{y}_{0:t-1})$ follows a normal distribution as follows:

$$q_\phi(\boldsymbol{\omega}_t^l \mid \boldsymbol{y}_{0:t-1}) = \mathcal{N}(\alpha_t^l \boldsymbol{\Omega}^l, (\alpha_t^l \boldsymbol{\Omega}^l)^2)$$
$$\rightleftharpoons \qquad\qquad (6)$$
$$\boldsymbol{\omega}_t^l = \alpha_t^l \boldsymbol{\Omega}^l (1 + \boldsymbol{\epsilon}) \quad , \boldsymbol{\epsilon} \sim \mathcal{N}(\mathbf{0}, \mathbb{I}),$$

where the square on $\boldsymbol{\Omega}^l$ is done component-wise, meaning weights in each layer are not correlated, and $\alpha_t^l$ are scalar positive dropout variational coefficients depending on $\boldsymbol{y}_{0:t-1}$ (and possibly $t$ depending on the specific applications). In practice, a network encoder $E_\psi$ is used to output the vector of dropouts coefficients for all layers $\boldsymbol{\alpha}_t = [\alpha_t^1, \ldots, \alpha_t^L] = E_\psi(\boldsymbol{y}_{0:t-1})$, with $\psi$ the encoder weights and the local-reparametrization trick is applied (Kingma et al., 2015) component-wise for linear layers. Thus, the variational parameter to optimize are $\phi = (\boldsymbol{\Omega}, \psi)$, with $\boldsymbol{\Omega} = \{\boldsymbol{\Omega}^1, \ldots, \boldsymbol{\Omega}^L\}$ static weights, and $\psi$ encoder weights.

**Aggregated Variational Posterior in Time Prior** To find a suitable prior for the model, we first observe that if the KL-divergence in eq. (4) is independent of $\boldsymbol{\Omega}$, then maximising the variational bound $\mathcal{L}$ with respect to $\boldsymbol{\Omega}$ for fixed $\boldsymbol{\alpha}_t$, is equivalent to maximise the expected log-likelihood $\log p(\boldsymbol{y}_t \mid \boldsymbol{\Omega}, \psi, \boldsymbol{y}_{0:t-1})$[2], where we made explicit the weight dependency. Thus, we look for a prior that allows us to have a KL-term independent of $\boldsymbol{\Omega}$.

Following the ideas of (Tomczak & Welling, 2018), we can find that the best prior for the lower bound in eq. (4) is given by (proof in Appendix A.2):

$$p(\boldsymbol{\omega}_t) = \int p(\boldsymbol{y}_{0:t-1}) q(\boldsymbol{\omega}_t \mid \boldsymbol{y}_{0:t-1}) \, d\boldsymbol{y}_{0:t-1}. \qquad (7)$$

We call this prior Temporal Variational Mixture of Posterior (tVAMP). This result is very similar to VAMP obtained in (Tomczak & Welling, 2018) but with a few differences: (i) VAMP is a prior on the latent variables, while tVAMP is a prior on the Bayesian Network weights; (ii) tVAMP is time-dependent, i.e. for each time $t$ the best prior is given by the aggregated posterior, while in (Tomczak & Welling, 2018) time dependency was not considered. Assuming the prior factorize similarly to the posterior, by carrying out the computations (see Appendix A.2), we obtain the following

---

[2]This is the same as optimising the network weights $\boldsymbol{\Omega}$ in standard deterministic networks.

prior indicating with $k$ the batch index:

$$p(\boldsymbol{\omega}_t^l) = \mathcal{N}(\beta_t^l \boldsymbol{\Omega}^l, (\gamma_t^l \boldsymbol{\Omega}^l)^2),$$

$$\beta_t^l = \frac{1}{N} \sum_{k=1}^{N} \alpha_t^l(\boldsymbol{y}_{0:t-1}^k),$$

$$\gamma_t^l = \sqrt{\frac{1}{N} \sum_{k=1}^{N} (\alpha_t^l(\boldsymbol{y}_{0:t-1}^k))^2} \quad \forall l, t. \tag{8}$$

Finally, for the given prior and posterior defined above, the KL-divergence is indeed independent of $\boldsymbol{\Omega}$ (proof in Appendix A.3) and reads:

$$D_{KL}\left[q_{\boldsymbol{\phi}}(\boldsymbol{\omega}_t \mid \boldsymbol{y}_{0:t-1}) \| p(\boldsymbol{\omega}_t)\right] =$$
$$\sum_{l=1}^{L} \frac{|\boldsymbol{\Omega}^l|}{2} \left[ \left(\frac{\alpha_t^l - \beta_t^l}{\gamma_t^l}\right)^2 + \left(\frac{\alpha_t^l}{\gamma_t^l}\right)^2 - 1 - 2\ln\frac{\alpha_t^l}{\gamma_t^l} \right] \tag{9}$$

where $|\boldsymbol{\Omega}^l|$ is the dimension of the weights in layer $l$.

### 3.4. Bayesian Neural PDE Solvers

Autoregressive Neural PDE Solvers map PDE states $\boldsymbol{y}_t$ to future states $\boldsymbol{y}_{t+1}$, given a specific initial state $\boldsymbol{y}_0$ (Brandstetter et al., 2022b). Those solvers are deterministic and compute the solution without providing estimates of uncertainty. BARNN can be used to convert any Autoregressive Neural PDE Solver into its Bayesian version by adopting a few steps. First, we model the joint weight-state distribution as:

$$p(\boldsymbol{y}_{0:T}, \boldsymbol{\omega}_{1:T}) = \prod_{t=1}^{T} p(\boldsymbol{y}_t \mid \boldsymbol{y}_{t-1}, \boldsymbol{\omega}_t) p(\boldsymbol{\omega}_t), \tag{10}$$

which is a special case of eq. (2) when markovianity of first order on the states is assumed. Then, we assume a Gaussian state distribution $p(\boldsymbol{y}_t \mid \boldsymbol{y}_{t-1}, \boldsymbol{\omega}_t) = \mathcal{N}(\text{NO}(\boldsymbol{y}_{t-1}; \boldsymbol{\omega}_t), \text{diag}(\boldsymbol{\sigma}_t^2))$, where the mean at time $t$ is obtained by applying the Autoregressive Neural PDE Solver to the state $\boldsymbol{y}_{t-1}$, while the standard deviation is not learned. Specifically, NO indicates *any* Neural PDE Solver architecture, with $\boldsymbol{\omega}_t$ its probabilistic weights, showing that the framework is independent of the specific architecture used. The probabilistic weights are obtained with the transformation $f(\boldsymbol{\Omega}, \boldsymbol{\alpha}_t, \boldsymbol{\epsilon})$ applied layerwise as explained in eq. (6), with the encoder $E_{\boldsymbol{\psi}}$ taking only one state as input (due to markovian updates). Learning is done by maximizing eq. (4), which, given the specific state distribution, becomes the minimization of the standard one-step MSE loss (Brandstetter et al., 2022a) commonly used to train Autoregressive Neural PDE Solvers, plus a Bayesian weight regularization term given by the negative KL divergence in eq. (9). The training algorithm is reported in Appendix D.

### 3.5. Bayesian Recurrent Neural Networks

Given a sequence $\boldsymbol{y}_0, \boldsymbol{y}_1, \ldots, \boldsymbol{y}_T$ a standard Recurrent Neural Network (RNN) (Bengio et al., 2000; Schmidhuber et al., 1997) computes eq. (1) by introducing hidden variables $\{\boldsymbol{h}_t\}_{t=0}^{t}$ that store[3] the input information up to the time $t$. Given these variables, the conditional probability is modelled as:

$$p(\boldsymbol{y}_t \mid \boldsymbol{y}_{t-1}, \ldots, \boldsymbol{y}_0) = \sigma(\boldsymbol{h}_{t-1})$$
$$\boldsymbol{h}_t = \text{NN}(\boldsymbol{y}_t, \boldsymbol{h}_{t-1}; \boldsymbol{\omega}), \tag{11}$$

where $\sigma$ is the softmax function, and NN is a neural network with deterministic weights $\boldsymbol{\omega}$ and a specific gate mechanism depending on the RNN structure. Importantly, the variable $\boldsymbol{h}_t$ contains historical information up to time $t$, i.e. knowledge of $\boldsymbol{y}_{t-1}, \ldots, \boldsymbol{y}_0$.

Extending RNNs to Bayesian RNNs is straightforward with BARNN. In particular, the states distribution becomes:

$$p(\boldsymbol{y}_t \mid \boldsymbol{y}_{t-1}, \ldots, \boldsymbol{y}_0, \boldsymbol{\omega}_t) = \sigma(\boldsymbol{h}_{t-1})$$
$$\boldsymbol{h}_t = \text{NN}(\boldsymbol{y}_t, \boldsymbol{h}_{t-1}; \boldsymbol{\omega}_t), \tag{12}$$

while the posterior distribution over the weights is obtained by conditioning on the hidden states:

$$q_{\boldsymbol{\psi}}(\boldsymbol{\omega}_t \mid \boldsymbol{y}_{t-1}, \ldots, \boldsymbol{y}_0) = q_{\boldsymbol{\psi}}(\boldsymbol{\omega}_t \mid \boldsymbol{y}_{t-1}, \boldsymbol{h}_{t-2}). \tag{13}$$

The probabilistic weights $\boldsymbol{\omega}_t$ are obtained again with the transformation $f(\boldsymbol{\Omega}, \boldsymbol{\alpha}_t, \boldsymbol{\epsilon})$ applied layerwise as explained in eq. (6), with the encoder $E_{\boldsymbol{\psi}}$ taking $\boldsymbol{y}_{t-1}, \boldsymbol{h}_{t-2}$ as input. Finally, learning is done by maximizing eq. (4), which, given the specific state distribution, becomes the minimization of the cross entropy loss (Bengio et al., 2000) commonly used in causal language modelling or next token prediction, plus the negative KL divergence in eq. (9). The training algorithm is reported in Appendix D.

## 4. Experiments

We demonstrate the effectiveness of BARNN by applying it to different tasks. To begin, we validate the tVAMP prior on a synthetic time series dataset and demonstrate its superiority over the widely used log-uniform prior. Then, we show that BARNN combined with Neural PDE Solvers (Bar-Sinai et al., 2019; Brandstetter et al., 2022b; Li et al., 2020) can solve PDEs and quantify the related uncertainty. Finally, we apply BARNN to RNNs for molecule unconditional generation (Segler et al., 2018; Özçelik et al., 2024) using the SMILES syntax and show stronger generation capabilities compared to existing methods. We make our code publicly available at https://github.com/dario-coscia/barnn.

---

[3]With the convention that $\boldsymbol{h}_{-1} = \boldsymbol{0}$.

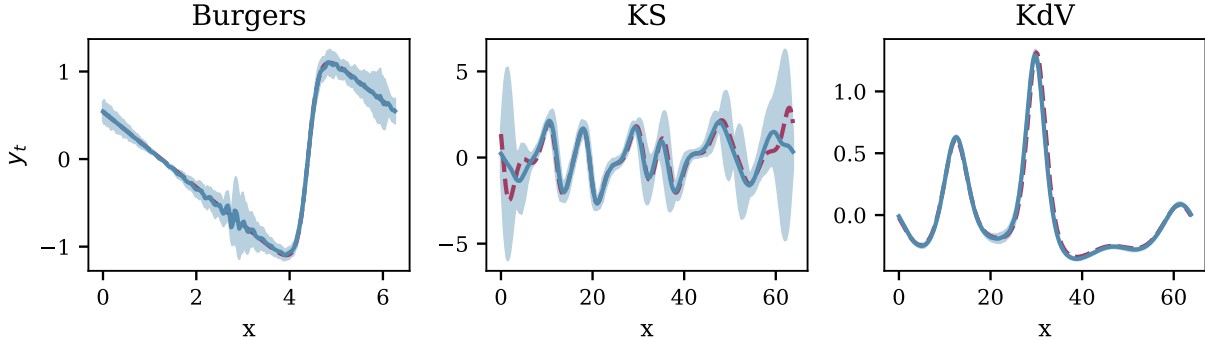

Figure 2: Prediction and uncertainty intervals for different PDEs at last time-step. The figure depicts the CFD solution (red dotted line) alongside the BARNN mean prediction (solid blue line) with ±3 std for uncertainty quantification (shaded blue area).

## 4.1. Time Series

We evaluate the performance of the proposed tVAMP prior on a synthetic time series forecasting task. The dataset is constructed by generating sequences of sinusoidal signals with varying frequencies and phases:

$$\begin{cases} x_t & = x_{t-1} + \frac{3\pi}{100}, \quad x_0 = 0 \\ y_t & = \frac{1}{5} \sum_{j=1}^{5} \sin(\alpha_i x_t + \beta_i), \end{cases} \quad (14)$$

with $\alpha_i \sim U[0.5, 1.5], \beta_i \sim U[0, 3\pi], t \in \{1, 2, \dots, 100\}$. This setup serves as a controlled environment to test the effect of prior choice on model uncertainty and forecasting accuracy. We generate 1024 trajectories for training, and 100 for testing. All models are trained using the same architecture, a 2-layer neural network of 64 units and Relu activation. The mean-square-error (MSE) loss is minimized for 1500 epochs using the Adam (Kingma & Ba, 2014) optimizer with learning rate $10^{-4}$ and weight decay $10^{-8}$.

**tVAMP Outperforms Log-Uniform Prior in Synthetic Forecasting Task** Table 1 presents an ablation study comparing different prior choices for BARNN on the synthetic time series dataset. We also compare BARNN against standard non-Bayesian MLP, and Monte Carlo Dropout (Dropout) (Gal & Ghahramani, 2016) with different levels of dropout probability. The BARNN model using the proposed tVAMP prior achieves the best performance across all metrics, outperforming both the log-uniform prior and standard baselines such as dropout and deterministic MLPs. Notably, tVAMP improves over the log-uniform prior, the negative log-likelihood from $-0.092$ to $-0.166$, indicating better-calibrated predictive uncertainty, while also achieving the lowest RMSE and ECE. These results demonstrate the effectiveness of the tVAMP prior in capturing uncertainty and enhancing forecasting accuracy in time series models.

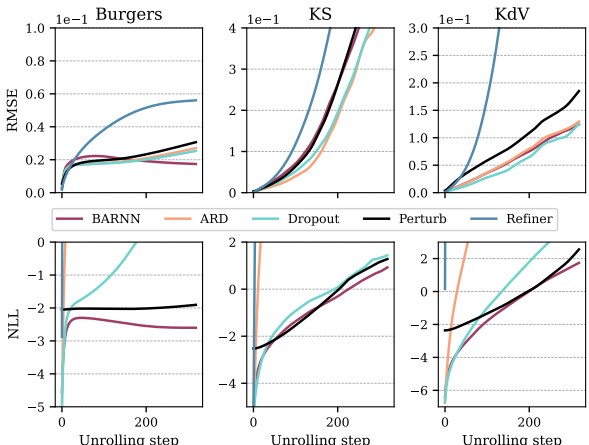

Figure 3: RMSE and NLL for different Neural Solvers predicting solutions to different PDEs.

## 4.2. PDE Modelling

PDE modelling consists of finding solutions to partial differential equations, generalising to unseen equations within a given family. We test BARNN on three famous PDEs, namely *Burgers*, *Kuramoto-Sivashinsky* (KS) and *Korteweg de Vries* (KdV) (Evans, 2022; Brandstetter et al., 2022a), and compare its performance against multiple UQ techniques for Neural PDE Solvers: Monte Carlo Dropout (Dropout) (Gal & Ghahramani, 2016), Variational Dropout with Empirical Bayes (ARD) (Kharitonov et al., 2018), which is a variation of Variational Dropout, and the recently proposed Input Perturbation (Lam et al., 2022) (Perturb), PDE Refiner (Lippe et al., 2024) (Refiner). A more detailed explanation of dataset generation, metrics, neural operator architectures, baselines and hyperparameters can be found in Appendix B, along with additional results.

Table 1: Ablation for different BARNN priors for the sinusoidal time-series forecasting dataset. The results report RMSE, NLL, and ECE statistics. Static method reports the RMSE obtained if the initial state is not propagated, while MLP is the base (non-Bayesian) architecture. BARNN uses the same MLP architecture, and it is ablated on different priors.

| Model | Prior | MSE ($\downarrow$) | NLL ($\downarrow$) | ECE ($\downarrow$) |
|---|---|---|---|---|
| Static | - | $0.490^{\pm 0.000}$ | - | - |
| MLP | - | $0.081^{\pm 0.011}$ | - | - |
| Dropout (p=0.5) | - | $0.072^{\pm 0.004}$ | $0.593^{\pm 0.461}$ | $0.084^{\pm 0.010}$ |
| Dropout (p=0.2) | - | $0.048^{\pm 0.004}$ | $-0.075^{\pm 0.004}$ | $0.068^{\pm 0.009}$ |
| BARNN | log-uniform | $0.045^{\pm 0.003}$ | $-0.092^{\pm 0.064}$ | $0.050^{\pm 0.016}$ |
| BARNN | tVAMP | $\mathbf{0.043}^{\pm 0.001}$ | $\mathbf{-0.166}^{\pm 0.019}$ | $\mathbf{0.049}^{\pm 0.008}$ |

Table 2: NLL and ECE for Burgers, KS and KdV PDEs for different Neural Solvers. The mean and std are computed for four different random weights initialization seeds. Solvers are unrolled for 320 steps, *break* is reported when the solver metric diverged.

| Model | NLL ($\downarrow$) | | |
|---|---|---|---|
| | *Burgers* | *KS* | *KdV* |
| BARNN | $\mathbf{-2.51}^{\pm 0.29}$ | $\mathbf{-0.87}^{\pm 0.52}$ | $\mathbf{-0.94}^{\pm 1.23}$ |
| ARD | $5.00^{\pm 7.11}$ | $10.7^{\pm 11.6}$ | break |
| Dropout | $0.06^{\pm 4.59}$ | $-0.48^{\pm 0.65}$ | $0.40^{\pm 4.57}$ |
| Perturb | $-2.01^{\pm 0.08}$ | $-0.65^{\pm 1.07}$ | $-0.37^{\pm 1.32}$ |
| Refiner | break | break | break |

| Model | ECE ($\downarrow$) | | |
|---|---|---|---|
| | *Burgers* | *KS* | *KdV* |
| BARNN | $\mathbf{0.05}^{\pm 0.03}$ | $\mathbf{0.04}^{\pm 0.03}$ | $\mathbf{0.10}^{\pm 0.01}$ |
| ARD | $0.26^{\pm 0.10}$ | $0.17^{\pm 0.01}$ | $0.19^{\pm 0.05}$ |
| Dropout | $0.17^{\pm 0.15}$ | $0.10^{\pm 0.01}$ | $0.14^{\pm 0.07}$ |
| Perturb | $0.16^{\pm 0.01}$ | $0.05^{\pm 0.02}$ | $0.11^{\pm 0.06}$ |
| Refiner | $0.30^{\pm 0.08}$ | $0.23^{\pm 0.02}$ | $0.34^{\pm 0.09}$ |

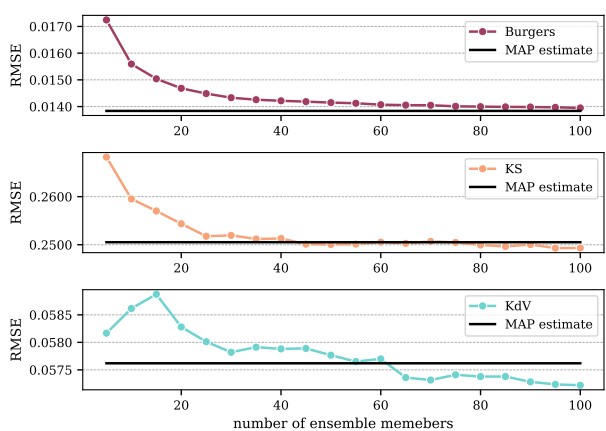

Figure 4: Variation of RMSE for increasing ensemble members in BARNN.

highlights that point-wise RMSE alone is insufficient for evaluating Neural PDE Solvers, especially in UQ scenarios, as low RMSE values can still mask incorrect uncertainty estimates from the solver.

**Confidence interval adaptability** Figure 2 illustrates the BARNN predictions alongside the 99.7% confidence intervals across different regions of the domain. Notably, BARNN dynamically adjusts the width of these confidence intervals based on the uncertainty present in each region. For example, in areas of high uncertainty, such as with the KS equation, the model appropriately broadens the intervals, reflecting its lower confidence in those predictions. Conversely, in regions where the model closely aligns with the true solution, as in the case of the KdV equation, the confidence intervals narrow considerably. This demonstrates BARNN's ability to flexibly adapt to varying levels of uncertainty, offering robust and reliable uncertainty quantification that aligns with the behaviour of the underlying PDEs.

**BARNN solves PDEs with calibrated uncertainties** We unroll the trained Neural Solvers for 320 steps and compute the negative log-likelihood (NLL) and the expected calibration error (ECE) of the predictions. NLL measures a balance between the accuracy of model prediction and the sharpness of the model's standard deviation, while the ECE measures how well the estimated probabilities match the observed probabilities. Table 2 reports the results, showing that BARNN can accurately solve PDEs with lower NLL and ECE compared to all existing methods. We found that, despite the models exhibiting similar root mean square errors (RMSE) , their negative log-likelihood varied significantly (Figure 3), driven by underestimated ensemble variance, which in turn caused a sharp increase in the NLL (as seen in models like Refiner and ARD). This finding

**Test number of ensemble members in BARNN** A desirable property of ensemble methods is that for a suffi-

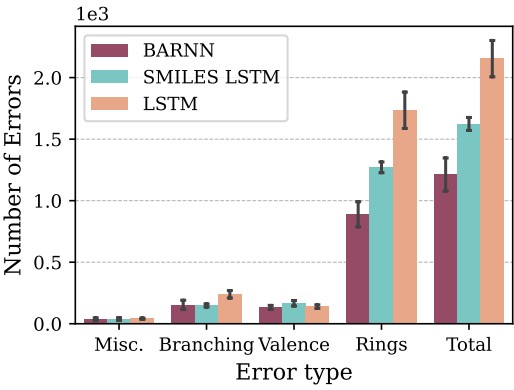

(a) Source of error for invalid generated SMILES.

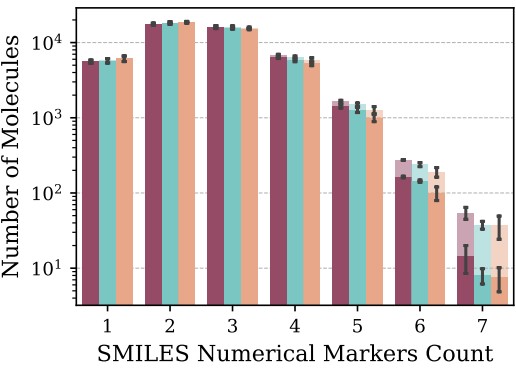

(b) Analysis of the impact of multiple rings in a SMILES string. Shaded colours indicate number of total SMILES (valid + invalid), while solid colours indicate only valid ones.

Figure 5: Error analysis for SMILES, mean and std shown for four different random weight initialization seeds.

ciently high number of ensemble members, the statistical moments (e.g. mean, std) converge. Figure 4 shows that with only $\simeq 30$ ensemble members BARNN provides convergent RMSE, emphasizing that accurate moments estimates can be obtained with just a few additional stochastic forward passes. In Appendix B.3 we also report ECE and NLL, which follow similar trends. Interestingly, if uncertainties are not required for a specific application, using the maximum a-posteriori estimate (MAP) for the network weights only requires a single forward pass to achieve a convergent RMSE, as depicted in Figure 4 . This makes the approach as fast as standard non-ensemble methods, which can be advantageous in practical applications.

## 4.3. Molecule Generation

Molecular generation involves creating new molecules with desired properties, which is difficult due to the vast chemical space needed to be explored, and long-range dependencies within molecular structures. We evaluate BARNN on the molecule generation task, by representing a molecule with the SMILES syntax and pre-train a recurrent neural network (LSTM gate-mechanism) (Schmidhuber et al., 1997) on the ChEMBL dataset from (Özçelik et al., 2024), which contains approximately 1.9 M SMILES. We test the BARNN LSTM model against standard LSTM (Schmidhuber et al., 1997), and the SMILES LSTM (Segler et al., 2018) which differ from the standard by applying Dropout to the LSTM input and hidden variables. Additional details on the dataset, metrics, hyperparameters, and baselines can be found in Appendix C.

**BARNN correctly learn the SMILES syntax** BARNN perfectly learns the SMILES syntax generating molecules with high validity, diversity, novelty and uniqueness compared to the baselines (Table 3). For validity, diversity and uniqueness BARNN archives 1% higher values than SMILES LSTM, and 2% than LSTM, while diversity is only slightly improved (0.1%) compared to the baselines.

**Long range dependencies in SMILES** We sampled $50K$ SMILES from the language models and for the invalid molecules examined the source of errors (Figure 5a). Interestingly, most of the errors language models commit are due to ring assignment, e.g. opening but not closing the ring. For ring assignments, BARNN outperforms the baselines, improving by 30% over SMILES LSTM and 50% over LSTM. In SMILES notation, ring closure bonds are represented by assigning matching digits to connected atoms, indicating where the ring opens and closes. A good language model must identify when the ring opens and remember to close it (with the same digit), a long-range dependency. This is analyzed by counting the numerical markers in each SMILES string and assessing validity (Figure 5b). As expected, SMILES strings with up to 4 rings show no errors, but validity decreases as the number of rings increases. BARNN outperforms baselines, generating more valid molecules, particularly for molecules with many rings, demonstrating superior handling of long-range dependencies.

**BARNN perfectly learns molecular properties** We test the ability of BARNN to learn the molecular properties of the training data by computing the Wasserstein distance $W_2$ between model-predicted properties and training properties. Following (Segler et al., 2018), we compute Molecular weight (MW), H-bond donors (HBDs), H-bond acceptors (HBAs), Rotable Bonds, LogP, and TPSA. Overall, the BARNN approach obtains the lowest Wasserstein distance for all molecular properties (Table 4), demonstrating an excellent ability to match the training data distribution. In particular, for molecular weight, which is the hardest prop-

Table 3: Designing drug-like molecules de novo with BARNN, mean and std shown for four different random weight initialization seeds.

| Model | Validity ($\uparrow$) | Diversity ($\uparrow$) | Novelty ($\uparrow$) | Uniqueness ($\uparrow$) |
|---|---|---|---|---|
| BARNN | $\mathbf{95.09}^{\pm 0.34}$ | $\mathbf{88.63}^{\pm 0.01}$ | $\mathbf{95.41}^{\pm 0.24}$ | $\mathbf{95.06}^{\pm 0.34}$ |
| SMILES LSTM | $94.60^{\pm 0.27}$ | $88.57^{\pm 0.09}$ | $94.25^{\pm 0.15}$ | $94.58^{\pm 0.28}$ |
| LSTM | $93.02^{\pm 0.33}$ | $88.57^{\pm 0.02}$ | $92.81^{\pm 0.34}$ | $92.14^{\pm 0.15}$ |

Table 4: Wasserstein distance between models' and training dataset's molecular properties distribution, mean and std shown for four different random weight initialization seeds.

| Model | $W_2$ ($\downarrow$) | | | | | |
|---|---|---|---|---|---|---|
| | MW | HBDs | HBAs | Rotable Bonds | LogP | TPSA |
| BARNN | $\mathbf{2.53}^{\pm 1.03}$ | $\mathbf{0.02}^{\pm 0.01}$ | $\mathbf{0.05}^{\pm 0.03}$ | $\mathbf{0.13}^{\pm 0.03}$ | $\mathbf{0.06}^{\pm 0.01}$ | $\mathbf{0.90}^{\pm 0.38}$ |
| SMILES LSTM | $10.4^{\pm 3.71}$ | $0.03^{\pm 0.02}$ | $0.15^{\pm 0.06}$ | $0.14^{\pm 0.05}$ | $0.11^{\pm 0.05}$ | $1.81^{\pm 0.97}$ |
| LSTM | $12.3^{\pm 2.91}$ | $0.03^{\pm 0.01}$ | $0.20^{\pm 0.07}$ | $\mathbf{0.13}^{\pm 0.07}$ | $0.12^{\pm 0.03}$ | $1.84^{\pm 1.04}$ |

erty to predict, BARNN obtains a substantial decrease in $W_2$ compared to the baselines. The results on molecular properties in Table 4, along with those on statistical properties in Table 3, demonstrate that BARNN not only more accurately reflects the dataset properties but also generates, on average, more statistically plausible molecules, significantly advancing the quality of molecular generation.

## 5. Conclusions

We have introduced BARNN, a scalable, calibrated and accurate methodology to turn any autoregressive or recurrent model to its Bayesian version, bridging the gap between autoregressive/ recurrent methods and Bayesian inference. BARNN creates a joint probabilistic model by evolving network weights along with states. We propose a novel variational lower bound for efficient training, extending Variational Dropout to compute dynamic network weights based on previous states, and introduce a temporal VAMP-prior to enhance calibration. We demonstrate BARNN's application in Autoregressive Neural PDE Solvers and molecule generation using RNNs. Experiments demonstrated that BARNN surpasses existing Neural PDE Solvers in accuracy and provides calibrated uncertainties. BARNN is also sharp, requiring only a few ensemble estimates and, if uncertainties are not needed, only one forward pass using the weights maximum a-posteriori. For molecules, BARNN generated molecules with higher validity, diversity, novelty and uniqueness compared to its non-Bayesian counterparts, and it also excels in learning long-range and molecular properties. BARNN is a very general and flexible framework that can be easily applied to other domains of DL, such as text, audio or time series. Several promising extensions for BARNN could also be explored. For example, developing more effective priors that incorporate prior knowledge could improve the model's performance. Another potential area of research is investigating local pruning at each autoregressive step, particularly for large over-parameterised networks such as LLMs. Finally, a deeper examination of the roles of epistemic and aleatory uncertainty over time represents an important direction for further study.

## Acknowledgments

D. Coscia, N. Demo and G. Rozza acknowledge the support provided by PRIN "FaReX - Full and Reduced order modeling of coupled systems: focus on non-matching methods and automatic learning" project, the European Research Council Executive Agency by the Consolidator Grant project AROMA-CFD "Advanced Reduced Order Methods with Applications in Computational Fluid Dynamics" - GA 681447, H2020-ERC CoG 2015 AROMA-CFD, PI G. Rozza, and the CINECA award under the ISCRA initiative, for high-performance computing resources and support availability. N. Demo and G. Rozza conducted this work within the research activities of the consortium iNEST (Interconnected North-East Innovation Ecosystem), Piano Nazionale di Ripresa e Resilienza (PNRR) – Missione 4 Componente 2, Investimento 1.5 – D.D. 1058 23/06/2022, ECS00000043, supported by the European Union's NextGenerationEU program.

## Impact Statement

This paper presents work whose goal is to advance the field of Machine Learning. There are many potential societal consequences of our work, none of which we feel must be specifically highlighted here.

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

# Supplementary Materials
# BARNN: Bayesian Autoregressive and Recurrent Neural Network

## Table of Contents

# A. Proofs and Derivations

This Appendix Section is devoted to the formal proofs introduced in the main text. In A.1 we prove the variational lower bound, while the t-VAMP proof is found in A.2. Finally, A.3 contains the derivation of KL-divergence between prior and posterior presented in 3.3.

## A.1. Variational Lower Bound proof

We aim to optimize the log-likelihood for the model:

$$p(\boldsymbol{y}_{0:T}, \boldsymbol{\omega}_{1:T}) = \prod_{t=1}^{T} p(\boldsymbol{y}_t \mid \boldsymbol{y}_{0:t-1}, \boldsymbol{\omega}_t) p(\boldsymbol{\omega}_t). \tag{15}$$

By defining the variational posterior:

$$q_\phi(\boldsymbol{\omega}_{1:T} \mid \boldsymbol{y}_{0:T}) = \prod_{t=1}^{T} q_\phi(\boldsymbol{\omega}_t \mid \boldsymbol{y}_{0:t-1}), \tag{16}$$

we obtain the following evidence lower bound:

$$
\begin{aligned}
\log p(\boldsymbol{y}_{0:T}) &= \log \int p(\boldsymbol{y}_{0:T}, \boldsymbol{\omega}_{1:T}) \, d\boldsymbol{\omega}_{1:T} \\
&= \log \int p(\boldsymbol{y}_{0:T}, \boldsymbol{\omega}_{1:T}) \frac{q_\phi(\boldsymbol{\omega}_{1:T} \mid \boldsymbol{y}_{0:T})}{q_\phi(\boldsymbol{\omega}_{1:T} \mid \boldsymbol{y}_{0:T})} \, d\boldsymbol{\omega}_{1:T} \\
&\geq \mathbb{E}_{\boldsymbol{\omega}_{1:T} \sim q_\phi(\boldsymbol{\omega}_{1:T}|\boldsymbol{y}_{0:T})} \left[ \log \frac{p(\boldsymbol{y}_{0:T}, \boldsymbol{\omega}_{1:T})}{q_\phi(\boldsymbol{\omega}_{1:T} \mid \boldsymbol{y}_{0:T})} \right] \\
&= \mathbb{E}_{\boldsymbol{\omega}_{1:T}} \left[ \log \prod_{t=1}^{T} \frac{p(\boldsymbol{y}_t \mid \boldsymbol{y}_{0:t-1}, \boldsymbol{\omega}_t) p(\boldsymbol{\omega}_t)}{q_\phi(\boldsymbol{\omega}_t \mid \boldsymbol{y}_{0:t-1})} \right] \\
&= \sum_{t=1}^{T} \mathbb{E}_{\boldsymbol{\omega}_t \sim q_\phi(\boldsymbol{\omega}_t|\boldsymbol{y}_{0:t-1})} \left[ \log \frac{p(\boldsymbol{y}_t \mid \boldsymbol{y}_{0:t-1}, \boldsymbol{\omega}_t) p(\boldsymbol{\omega}_t)}{q_\phi(\boldsymbol{\omega}_t \mid \boldsymbol{y}_{0:t-1})} \right] \\
&= \sum_{t=1}^{T} \left\{ \mathbb{E}_{\boldsymbol{\omega}_t \sim q_\phi(\boldsymbol{\omega}_t|\boldsymbol{y}_{0:t-1})} \left[ \log p(\boldsymbol{y}_t \mid \boldsymbol{y}_{0:t-1}, \boldsymbol{\omega}_t) \right] - D_{KL} \left[ q_\phi(\boldsymbol{\omega}_t \mid \boldsymbol{y}_{0:t-1}) \| p(\boldsymbol{\omega}_t) \right] \right\} \\
&= \sum_{t=1}^{T} \mathcal{L}_{\text{cumulative}}(\boldsymbol{\phi}, t)
\end{aligned}
\tag{17}
$$

Interestingly, by being Bayesian, we ended up having a sum of evidence lower bounds, which tells us that to maximise the model likelihood we need to maximise the *cumulative evidence lower bound*. For performing scalable training we approximate the found evidence lower bound as usually done in causal language modelling or next token prediction by:

$$\mathcal{L}(\boldsymbol{\phi}) \approx \mathbb{E}_{t \sim \mathcal{U}[1,T]} \left[ \mathcal{L}_{\text{cumulative}}(\boldsymbol{\phi}, t) \right]. \tag{18}$$

## A.2. Variational Mixture of Posterior in Time proof

We want to find the best prior that maximises the evidence lower bound:

$$\mathcal{L}(\boldsymbol{\phi}) = \mathbb{E}_{\boldsymbol{\omega}_{1:T} \sim q_\phi(\boldsymbol{\omega}_{1:T}|\boldsymbol{y}_{0:T})} \left[ \log \frac{p(\boldsymbol{y}_{0:T}, \boldsymbol{\omega}_{1:T})}{q_\phi(\boldsymbol{\omega}_{1:T} \mid \boldsymbol{y}_{0:T})} \right]$$

then:

$$
\begin{aligned}
p^*(\boldsymbol{\omega}_{1:T}) &= \arg\max_{p(\boldsymbol{\omega}_{1:T})} \int \mathcal{L}(\boldsymbol{\phi}) p(\boldsymbol{y}_{0:T})\, d\boldsymbol{y}_{0:T} \\
&= \arg\max_{p(\boldsymbol{\omega}_{1:T})} \int p(\boldsymbol{y}_{0:T}) q(\boldsymbol{\omega}_{1:T} \mid \boldsymbol{y}_{0:T}) \log p(\boldsymbol{\omega}_{1:T})\, d\boldsymbol{y}_{0:T} d\boldsymbol{\omega}_{1:T} \\
&= \arg\max_{p(\boldsymbol{\omega}_{1:T})} \int \left[ \int p(\boldsymbol{y}_{0:T}) q(\boldsymbol{\omega}_{1:T} \mid \boldsymbol{y}_{0:T})\, d\boldsymbol{y}_{0:T} \right] \log p(\boldsymbol{\omega}_{1:T}) d\boldsymbol{\omega}_{1:T} \\
&= \arg\max_{p(\boldsymbol{\omega}_{1:T})} - H\left[ \int p(\boldsymbol{y}_{0:T}) q(\boldsymbol{\omega}_{1:T} \mid \boldsymbol{y}_{0:T})\, d\boldsymbol{y}_{0:T} \mid p(\boldsymbol{\omega}_{1:T}) \right] \\
&= \int p(\boldsymbol{y}_{0:T}) q(\boldsymbol{\omega}_{1:T} \mid \boldsymbol{y}_{0:T})\, d\boldsymbol{y}_{0:T}
\end{aligned}
\tag{19}
$$

where $H$ is the cross entropy and has a maximum when the two distributions match. In particular, for each time $t$:

$$
p^*(\boldsymbol{\omega}_t) = \int p^*(\boldsymbol{\omega}_{1:T})\, d\boldsymbol{\omega}_{\neq t} = \int p(\boldsymbol{y}_{0:T}) q(\boldsymbol{\omega}_{1:T} \mid \boldsymbol{y}_{0:T})\, d\boldsymbol{y}_{0:T}\, d\boldsymbol{\omega}_{\neq t} = \int p(\boldsymbol{y}_{0:t-1}) q(\boldsymbol{\omega}_t \mid \boldsymbol{y}_{0:t-1})\, d\boldsymbol{y}_{0:t-1}.
$$

This result is very similar to the one obtained in (Tomczak & Welling, 2018) but with a few differences: (i) (Tomczak & Welling, 2018) is a prior on the latent variables, here it is a prior on the Bayesian Network weights; (ii) the presented prior is time-dependent, i.e. for each time $t$ the best prior is given by the aggregated posterior, while in (Tomczak & Welling, 2018) time dependency was not considered.

We approximate the aggregated distribution assuming it factorizes similarly to the variational posterior:

$$
p^*(\boldsymbol{\omega}_t) \approx \prod_{l=1}^{L} \prod_{i_l j_l} \int p(\boldsymbol{y}_{0:t-1}) \mathcal{N}(\alpha_t^l \Omega_{i_l j_l}^l, (\alpha_t^l \Omega_{i_l j_l}^l)^2) d\boldsymbol{y}_{0:t-1} = \prod_{l=1}^{L} \prod_{i_l j_l} \mathcal{N}(\beta_t^l \Omega_{i_l j_l}^l, (\gamma_t^l \Omega_{i_l j_l}^l)^2),
\tag{20}
$$

where, we can compute $\beta_t^l$, and $\gamma_l^t$ as functions of $\alpha_l^t$:

$$
\begin{aligned}
\beta_t^l \Omega_{i_l j_l}^l &= \mathbb{E}_{\omega_{i_l j_l}^l}[\omega_{i_l j_l}^l] \\
&= \int p(\boldsymbol{y}_{0:t-1}) \left[ \int \omega_{i_l j_l}^l \mathcal{N}(\alpha_t^l \Omega_{i_l j_l}^l, (\alpha_t^l \Omega_{i_l j_l}^l)^2) d\omega_{i_l j_l}^l \right] d\boldsymbol{y}_{0:t-1} \\
&= \Omega_{i_l j_l}^l \int \alpha_t^l(\boldsymbol{y}_{0:t-1}) p(\boldsymbol{y}_{0:t-1}) d\boldsymbol{y}_{0:t-1} \approx \Omega_{i_l j_l}^l \frac{1}{N} \sum_{k=1}^{N} \alpha_t^l(\boldsymbol{y}_{0:t-1}^k)
\end{aligned}
\tag{21}
$$

and,

$$
\begin{aligned}
(\gamma_t^l \Omega_{i_l j_l}^l)^2 &= \mathbb{V}ar_{\omega_{i_l j_l}^l}[\omega_{i_l j_l}^l] \\
&= \int p(\boldsymbol{y}_{0:t-1}) \left[ \int (\omega_{i_l j_l}^l - \alpha_t^l \Omega_{i_l j_l}^l)^2 \mathcal{N}(\alpha_t^l \Omega_{i_l j_l}^l, (\alpha_t^l \Omega_{i_l j_l}^l)^2) d\omega_{i_l j_l}^l \right] d\boldsymbol{y}_{0:t-1} \\
&= (\Omega_{i_l j_l}^l)^2 \int (\alpha_t^l(\boldsymbol{y}_{0:t-1}))^2 p(\boldsymbol{y}_{0:t-1}) d\boldsymbol{y}_{0:t-1} \approx (\Omega_{i_l j_l}^l)^2 \frac{1}{N} \sum_{k=1}^{N} (\alpha_t^l(\boldsymbol{y}_{0:t-1}^k))^2.
\end{aligned}
\tag{22}
$$

In summary,

$$
\beta_t^l = \frac{1}{N} \sum_{k=1}^{N} \alpha_t^l(\boldsymbol{y}_{t-1}^k, t), \quad \gamma_t^l = \sqrt{\frac{1}{N} \sum_{k=1}^{N} (\alpha_t^l(\boldsymbol{y}_{t-1}^k, t))^2} \quad \forall l, t.
\tag{23}
$$

### A.3. KL-divergence proof

We want to compute $D_{KL}\left[q_\phi(\boldsymbol{\omega}_t \mid \boldsymbol{y}_{0:t-1})\|p(\boldsymbol{\omega}_t)\right]$. Due to factorization on the layers, the KL divergence reads:

$$
\begin{aligned}
D_{KL}\left[q_\phi(\boldsymbol{\omega}_t \mid \boldsymbol{y}_{0:t-1})\|p(\boldsymbol{\omega}_t)\right] &= \sum_{l=1}^{L} \mathbb{E}_{\boldsymbol{\omega}_t^l \sim q_\phi(\boldsymbol{\omega}_t^l \mid \boldsymbol{y}_{0:t-1})}\left[\log\left(\frac{q_\phi(\boldsymbol{\omega}_t^l \mid \boldsymbol{y}_{0:t-1})}{p(\boldsymbol{\omega}_t^l)}\right)\right] \\
&= \sum_{l=1}^{L} D_{KL}\left[q_\phi(\boldsymbol{\omega}_t^l \mid \boldsymbol{y}_{0:t-1})\|p(\boldsymbol{\omega}_t^l)\right].
\end{aligned}
\tag{24}
$$

We now expand the KL divergence $D_{KL}\left[q_\phi(\boldsymbol{\omega}_t^l \mid \boldsymbol{y}_{0:t-1})\|p(\boldsymbol{\omega}_t^l)\right]$ for each layer $l$. Let $\omega_{ij;t}^l$ representing the matrix entries of $\boldsymbol{\omega}_t^l$, then:

$$
D_{KL}\left[q_\phi(\boldsymbol{\omega}_t^l \mid \boldsymbol{y}_{0:t-1})\|p(\boldsymbol{\omega}_t^l)\right] = \sum_{i_l,j_l} \mathbb{E}_{\omega_{ij;t}^l \sim q_\phi(\omega_{ij;t}^l \mid \boldsymbol{y}_{0:t-1})}\left[\log\left(\frac{q_\phi(\omega_{ij;t}^l \mid \boldsymbol{y}_{0:t-1})}{p(\omega_{ij;t}^l)}\right)\right].
\tag{25}
$$

We model $q_\phi(\omega_{ij;t}^l \mid \boldsymbol{y}_{0:t-1}) = \mathcal{N}(\alpha_t^l \Omega_{ij}^l, (\alpha_t^l \Omega_{ij}^l)^2)$, and $p(\omega_{ij;t}^l) = \mathcal{N}(\beta_t^l \Omega_{ij}^l, (\gamma_t^l \Omega_{ij}^l)^2)$, where $\alpha_t^l, \beta_t^l, \gamma_t^l$ are the dropout rates (see in the main text). Then eq. (25) becomes:

$$
\begin{aligned}
D_{KL}\left[q_\phi(\boldsymbol{\omega}_t^l \mid \boldsymbol{y}_{0:t-1})\|p(\boldsymbol{\omega}_t^l)\right] &= \sum_{i_l,j_l} \frac{1}{2}\left[\frac{(\alpha_t^l \Omega_{ij}^l - \beta_t^l \Omega_{ij}^l)^2}{(\gamma^l \Omega_{ij}^l)^2} + \left(\frac{(\alpha_t^l \Omega_{ij}^l)^2}{(\gamma_t^l \Omega_{ij}^l)^2}\right) - 1 - \ln\left(\frac{(\alpha_t^l \Omega_{ij}^l)^2}{(\gamma_t^l \Omega_{ij}^l)^2}\right)\right] \\
&= \sum_{i_l,j_l} \frac{1}{2}\left[\left(\frac{\alpha_t^l - \beta_t^l}{\gamma_t^l}\right)^2 + \left(\frac{\alpha_t^l}{\gamma_t^l}\right)^2 - 1 - 2\ln\left(\frac{\alpha_t^l}{\gamma_t^l}\right)\right].
\end{aligned}
\tag{26}
$$

In the last step, we used the fact that the dropout rates are positive. Combining eq. (26) with eq. (24) concludes the proof. Notice that this KL divergence is independent on $\Omega_{ij}^l \; \forall i, j, l$.

### A.4. Predictive Distribution and Uncertainty Estimates

This subsection shows how to use BARNN for inference, and uncertainty quantification. In the inference case, we are interested in sampling from previous states $\boldsymbol{y}_0, \ldots, \boldsymbol{y}_{t-1}$ the causally consecutive one $\boldsymbol{y}_t$, while in UQ we aim to model epistemic (model weight uncertainty) and/or aleatory uncertainty (data's inherent randomness).

**Predictive Distribution**   The model *predictive distribution* $p(\boldsymbol{y}_t \mid \boldsymbol{y}_{t-1}, \ldots, \boldsymbol{y}_0)$ is obtained by marginalizing over the weights $\boldsymbol{\omega}_t$ sampled from the posterior distribution:

$$
p(\boldsymbol{y}_t \mid \boldsymbol{y}_{t-1}, \ldots, \boldsymbol{y}_0) = \mathbb{E}_{\boldsymbol{\omega}_t \sim q_\phi}\left[p(\boldsymbol{y}_t \mid \boldsymbol{y}_{0:t-1}, \boldsymbol{\omega}_t) q_\phi(\boldsymbol{\omega}_t \mid \boldsymbol{y}_{0:t-1})\right].
\tag{27}
$$

In practical terms, this involves executing multiple stochastic forward passes through the neural networks and then averaging the resulting outputs (Gal & Ghahramani, 2016; Louizos & Welling, 2017).

**Uncertainty Estimates**   The predictive distribution allows for a clear variance decomposition using the law of total variance (Depeweg et al., 2018):

$$
\boldsymbol{\sigma}_t^2 = \underbrace{\boldsymbol{\sigma}_{\boldsymbol{\omega}_t}^2(\mathbb{E}[\boldsymbol{y}_t \mid \boldsymbol{y}_{0:t-1}, \boldsymbol{\omega}_t])}_{\text{epistemic uncertainty}} + \underbrace{\mathbb{E}_{\boldsymbol{\omega}_t}[\boldsymbol{\sigma}^2(\boldsymbol{y}_t \mid \boldsymbol{y}_{0:t-1}, \boldsymbol{\omega}_t)]}_{\text{aleatoric uncertainty}},
\tag{28}
$$

with $\boldsymbol{\omega}_t \sim q_\phi(\boldsymbol{\omega}_t \mid \boldsymbol{y}_{0:t-1})$, and $\mathbb{E}[\boldsymbol{y}_t \mid \boldsymbol{y}_{0:t-1}, \boldsymbol{\omega}_t], \boldsymbol{\sigma}^2(\boldsymbol{y}_t \mid \boldsymbol{y}_{0:t-1}, \boldsymbol{\omega}_t)$ the mean and variance of the distribution $p(\boldsymbol{y}_t \mid \boldsymbol{y}_{t-1}, \ldots, \boldsymbol{y}_0, \boldsymbol{\omega}_t)$ respectively.

**Monte Carlo Estimates**   Samples from eq. (27), and uncertainties from eq. (28) can be computed by Monte Carlo estimates. Assume, for a fixed initial state $\boldsymbol{y}_0$, to unroll an autoregressive or recurrent network for a full trajectory $i$-times,

with $i = (1, \ldots, D)$, each time sampling different $\boldsymbol{\omega}_t$. Let $\boldsymbol{\mu}_{t;i}, \boldsymbol{\sigma}^2_{t;i}$ the mean and variance of $p(\boldsymbol{y}_t \mid \boldsymbol{y}_{0:t-1}, \boldsymbol{\omega}_t)$ for the $i$-sample of $\boldsymbol{\omega}_t$. Then, the following unbiased Monte Carlo estimates can be used:

$$\boldsymbol{\mu}_t = \frac{1}{D} \sum_{i=1}^{D} \boldsymbol{\mu}_{t;i} \ , \qquad \boldsymbol{\sigma}^2_t = \underbrace{\frac{1}{D} \sum_{i=1}^{D} \boldsymbol{\mu}^2_{t;i} - \boldsymbol{\mu}^2_t}_{\text{epistemic uncertainty}} + \underbrace{\frac{1}{D} \sum_{i=1}^{D} \boldsymbol{\sigma}^2_{t;i}}_{\text{aleatoric uncertainty}} \ . \tag{29}$$

### A.5. Connection to not Bayesian Autoregressive and Recurrent Networks

In this subsection we want to derive a connection between the variational lower bound in eq. (4), and the standard log-likelihood commonly used for training autoregressive networks. We start by rewriting eq. (4):

$$\mathcal{L}(\boldsymbol{\phi}) = \mathbb{E}_{t \sim \mathcal{U}[1,T]} \left[ \mathbb{E}_{\boldsymbol{\omega}_t \sim q_{\boldsymbol{\phi}}(\boldsymbol{\omega}_t \mid \boldsymbol{y}_{0:t-1})} \left[ \log p(\boldsymbol{y}_t \mid \boldsymbol{y}_{0:t-1}, \boldsymbol{\omega}_t) \right] - D_{KL} \left[ q_{\boldsymbol{\phi}}(\boldsymbol{\omega}_t \mid \boldsymbol{y}_{0:t-1}) \| p(\boldsymbol{\omega}_t) \right] \right] . \tag{30}$$

Now we make the following assumptions:

(a.1) $\boldsymbol{\omega}_t = \boldsymbol{\omega} \quad \forall t = (1, 2, \ldots)$,

(a.2) $q(\boldsymbol{\omega} \mid \boldsymbol{y}_{0:t-1}) = p(\boldsymbol{\omega}) = \delta(\boldsymbol{\omega} - \boldsymbol{\Omega}) \quad \forall t = (1, 2, \ldots)$

Assumption (a.1) ensures that weights do not change over time (*static weights*). Assumption (a.2) fixes the weights to a single value $\boldsymbol{\Omega}$ (*deterministic weights*).

Under assumption (a.2) the KL-divergence is zero, and with assumption (a.1) the expectation reduces to a single value; thus we are left with the following:

$$\mathcal{L} = \mathbb{E}_{t \sim \mathcal{U}[1,T]} \left[ \log p(\boldsymbol{y}_t \mid \boldsymbol{y}_{0:t-1}, \boldsymbol{\Omega}) \right] \tag{31}$$

which is the standard log-likelihood optimized in autoregressive networks.

# B. BARNN PDEs Application

This section shows how BARNN can be used to build Bayesian Neural PDE Solvers, provides all the details for reproducing the experiments, and shows additional results.

## B.1. Data Generation and Metrics

**Data Generation:** We focus on PDEs modelling evolution equations, although our method can be applied to a vast range of time-dependent differential equations. Specifically we consider three famous types of PDEs, commonly used as benchmark for Neural PDE Solvers (Brandstetter et al., 2022a;b; Bar-Sinai et al., 2019):

$$
\begin{aligned}
\textit{Burgers} && \partial_t u + u\partial_x u - \nu\partial_{xx} u = 0 \\
\textit{Kuramoto Sivashinsky} && \partial_t u + u\partial_x u + \partial_{xx} u + \partial_{xxxx} u = 0 \\
\textit{Korteweg de Vries} && \partial_t u + u\partial_x u + \partial_{xxx} u = 0
\end{aligned}
\tag{32}
$$

where $\nu > 0$ is the viscosity coefficient. We follow the data generation setup of (Brandstetter et al., 2022a), applying periodic boundary conditions, and sampling the initial conditions $u^0$ from a distribution over truncated Fourier series $u^0(x) = \sum_{k=1}^{10} A_k \sin(2\pi l_k \frac{x}{L} + \eta_k)$, where $\{A_k, \eta_k, l_k\}_k$ are random coefficients as in (Brandstetter et al., 2022a) The space-time discretization parameters are reported in Table 5.

Table 5: PDE parameters setup. The discretization in time and space is indicated by $n_t$ and $n_x$ respectively, $t_{\max}$ represents the simulation physical time, and $L$ is the domain length.

|       | PDE | $n_t$ | $n_x$ | $t_{\max}$ | $L$ |
|-------|-----|-------|-------|------------|-----|
|       | *Burgers* | 0.1 | 0.25 | 14 | $2\pi$ |
| Train | *KS* | 0.1 | 0.25 | 14 | 64 |
|       | *KdV* | 0.1 | 0.25 | 14 | 64 |
|       | *Burgers* | 0.1 | 0.25 | 32 | $2\pi$ |
| Test  | *KS* | 0.1 | 0.25 | 32 | 64 |
|       | *KdV* | 0.1 | 0.25 | 32 | 64 |

**Metrics:** To evaluate the performance of the probabilistic solvers, we focus on different metrics:

1. **Root Mean Square Error** (RMSE) (Brandstetter et al., 2022b): measures the match of the ensemble prediction means and true values

2. **Negative Log-Likelihood** (NLL): represents the trade-off between low standard deviations and error terms, where the latter are between ensemble prediction means and true values

3. **Expected Calibration Error** (ECE): measures how well the estimated probabilities match the observed probabilities

Let $\mu_t, \sigma_t$ the ensemble mean and variance predicted by the probabilistic Neural Solver for different times $t$, $\mathbb{Q}$ the Gaussian quantile function of $p$, then:

$$
\begin{aligned}
\text{RMSE} &= \sqrt{\frac{1}{n_x n_t} \sum_{x,t} [y_t(x) - \mu_t(x)]^2} \\
\text{NLL} &= \frac{1}{2n_x n_t} \sum_{x,t} \left[ \log(2\pi\sigma_t^2(x)) + \frac{[u_t(x) - \mu_t(x)]^2}{\sigma_t^2(x)} \right] \\
\text{ECE} &= \mathbb{E}_{p\sim U(0,1)}\left[ |p - p_{obs}| \right] \quad, p_{obs} = \frac{1}{n_x n_t} \sum_{x,t} \mathbb{I}\left\{ y_t(x) \leq \mathbb{Q}_p[\mu_t(x), \sigma_t^2(x)] \right\}
\end{aligned}
\tag{33}
$$

## B.2. Model Architectures, Hyperparameters, and Computational Costs

All models were trained for 7000 epochs using Adam with $5 \cdot 10^{-4}$ of learning rate and $10^{-8}$ of weight decay for regularization. Computations were performed on a single Quadro RTX 4000 GPU with 8-GB of memory and requires approximately one day to train all models. Unless otherwise stated, the mean and std of the predictions are computed using 100 ensemble members for all models, and models are unrolled for 320 steps in inference.

We perform all experiments with a Fourier Neural Operator (Li et al., 2020) which is a standard baseline for Neural PDE Solvers. The Neural Operator is composed of 8 layers of size 64 with 32 modes and swish activation (Li et al., 2020; Brandstetter et al., 2022a) for a total of approximately 1 million trainable parameters. We found that using batch normalization worsens all the uncertainty estimates. We used one linear layer to lift the input and one to project the output. For dropout models (Dropout and ARD Dropout), we found that applying dropout only on the linear layer of the Fourier layer and on the projection layer gave the best uncertainties, we used this also for BARNN for a fair comparison. In the following, we report the specific hyperparameters for each model.

**Dropout** We used a dropout rate of $0.2$, applying it only to the Fourier linear layers. We also tested other dropout rates $0.5, 0.8$, but they worsened the uncertainty. In Table 6 we report an ablation study for different dropout rates, due to the computational burden we run the study only for one seed.

Table 6: Results for different PDEs and Dropout rates on the validation set for 140 steps unrolling. Results report RMSE, NLL, and ECE statistics.

| PDE | Dropout Rate | RMSE ($\downarrow$) | NLL ($\downarrow$) | ECE ($\downarrow$) |
|---|---|---|---|---|
| | 0.2 | 0.014 | -3.070 | 0.022 |
| *Burgers* | 0.5 | 0.030 | -1.779 | 0.077 |
| | 0.8 | 0.055 | -1.562 | 0.030 |
| | 0.2 | 0.044 | -0.416 | 0.120 |
| *KS* | 0.5 | 0.066 | -1.247 | 0.065 |
| | 0.8 | 0.183 | 0.911 | 0.127 |
| | 0.2 | 0.017 | -3.350 | 0.135 |
| *KdV* | 0.5 | 0.040 | -2.280 | 0.078 |
| | 0.8 | 0.080 | -1.310 | 0.097 |

**Input Perturbation** We employed the standard FNO discussed above and perturbed the input with Gaussian noise during training (Lam et al., 2023), $\boldsymbol{u}_t^{\text{perturb}} = \boldsymbol{u}_t + \delta \max |\boldsymbol{u}_t| \boldsymbol{\epsilon}$, with $\boldsymbol{\epsilon} \sim N(\boldsymbol{0}, \mathbb{I})$ and $\delta = 0.01$. In Table 7 we report an ablation study for different perturbation scales, due to the computational burden we run the study only for one seed.

**PDE Refiner** We implemented our version of PDE Refiner, following the pseudocode provided in the paper (Lippe et al., 2024). We found the model to be very sensitive to the choice of the refinement steps $R$ and minimum variance $\sigma_{\min}$ hyperparameters, in accordance with previous studies (Kohl et al., 2024). For the PDE experiments we used $R = 2, \sigma_{\min} = 2 \cdot 10^{-6}$. In Table 8 we report an ablation study for different perturbation scales, due to the computational burden we run the study only for one seed.

**ARD-Dropout** The model does not have hyperparameters, we follow (Kharitonov et al., 2018) for the implementation. We used a global dropout rate to have a fair comparison with standard Dropout, and we applied ARD-Dropout only to the Fourier layers.

**BARNN** Similarly to Dropout and ARD-Dropout, we applied the dropout mask only on the Fourier fully connected layers. The dropout coefficients $\boldsymbol{\alpha}_t$ are given by a specific encoder $E_{\boldsymbol{\psi}}$. In our implementation, the encoder takes as input the state $\boldsymbol{y}_t$ and the time variable $t$. We map the state channel to a fixed dimension of 16 using one linear layer. Consequently, we map the time variable using a sinusoidal positional encoding with period $T = 1000$ to scale and shift the state. The resulting state embedding is mapped to the output shape (number of variational layers) with four other linear layers of

Table 7: Results for different PDEs and Input Perturbations scales $\delta$ on the validation set for $140$ steps unrolling. Results report RMSE, NLL, and ECE statistics.

| PDE | Perturbation | RMSE ($\downarrow$) | NLL ($\downarrow$) | ECE ($\downarrow$) |
|---|---|---|---|---|
| | 0.001 | 0.017 | 1.583 | 0.366 |
| *Burgers* | 0.01 | 0.014 | -2.085 | 0.181 |
| | 0.1 | 0.093 | 0.218 | 0.193 |
| | 0.001 | 0.015 | -1.048 | 0.188 |
| *KS* | 0.01 | 0.041 | -1.942 | 0.110 |
| | 0.1 | 0.404 | 0.413 | 0.078 |
| | 0.001 | 0.014 | -0.641 | 0.231 |
| *KdV* | 0.01 | 0.027 | -2.032 | 0.111 |
| | 0.1 | 0.155 | 0.115 | 0.189 |

Table 8: Results for different PDEs and Refiner Hyperparameters on the validation set for $140$ steps unrolling. Results report RMSE, NLL, and ECE statistics.

| PDE | $\sigma_{\min}$ | $R$ | RMSE ($\downarrow$) | NLL ($\downarrow$) | ECE ($\downarrow$) |
|---|---|---|---|---|---|
| | $2 \cdot 10^{-6}$ | 2 | 0.019 | 44.18 | 0.216 |
| | $2 \cdot 10^{-6}$ | 3 | 0.106 | 184.4 | 0.547 |
| | $2 \cdot 10^{-6}$ | 4 | 0.021 | 16.64 | 0.196 |
| | $2 \cdot 10^{-7}$ | 2 | 0.014 | 86.24 | 0.248 |
| *Burgers* | $2 \cdot 10^{-7}$ | 3 | 0.026 | 240.4 | 0.239 |
| | $2 \cdot 10^{-7}$ | 4 | 0.023 | 100.3 | 0.367 |
| | $2 \cdot 10^{-6}$ | 2 | 0.027 | 6.089 | 0.167 |
| | $2 \cdot 10^{-6}$ | 3 | 0.081 | 76.76 | 0.232 |
| | $2 \cdot 10^{-6}$ | 4 | 0.041 | 8.456 | 0.179 |
| | $2 \cdot 10^{-7}$ | 2 | 0.024 | 9.330 | 0.189 |
| *KS* | $2 \cdot 10^{-7}$ | 3 | 0.025 | 6.977 | 0.183 |
| | $2 \cdot 10^{-7}$ | 4 | 0.037 | 6.250 | 0.175 |
| | $2 \cdot 10^{-6}$ | 2 | 0.011 | 10.17 | 0.402 |
| | $2 \cdot 10^{-6}$ | 3 | 0.031 | 72.10 | 0.343 |
| | $2 \cdot 10^{-6}$ | 4 | 0.022 | 39.90 | 0.285 |
| | $2 \cdot 10^{-7}$ | 2 | 0.043 | 86.74 | 0.356 |
| *KdV* | $2 \cdot 10^{-7}$ | 3 | 0.042 | 192.5 | 0.456 |
| | $2 \cdot 10^{-7}$ | 4 | 0.042 | 92.33 | 0.321 |

size 16 interleaved with swish activation and a final sigmoid activation to return the dropout probabilities $p_t$. The dropout rates can be obtained by computing $p_t/(1 - p_t)$. The built encoder contains less than 50K parameters, thus a tiny network compared to modern Neural Solver networks, not significantly affecting the training time.

### B.3. Additional Results

**Comparison of Model Accuracies** Table 9 presents the RMSE results for the analysed PDEs across various models. The dropout-based methods (BARNN, ARD, and Dropout) consistently show the best performance, indicating that Bayesian approaches can be advantageous not only for uncertainty quantification but also for pointwise accuracy. The differences in RMSE are relatively small among the top-performing models, suggesting that they generally perform at a similar

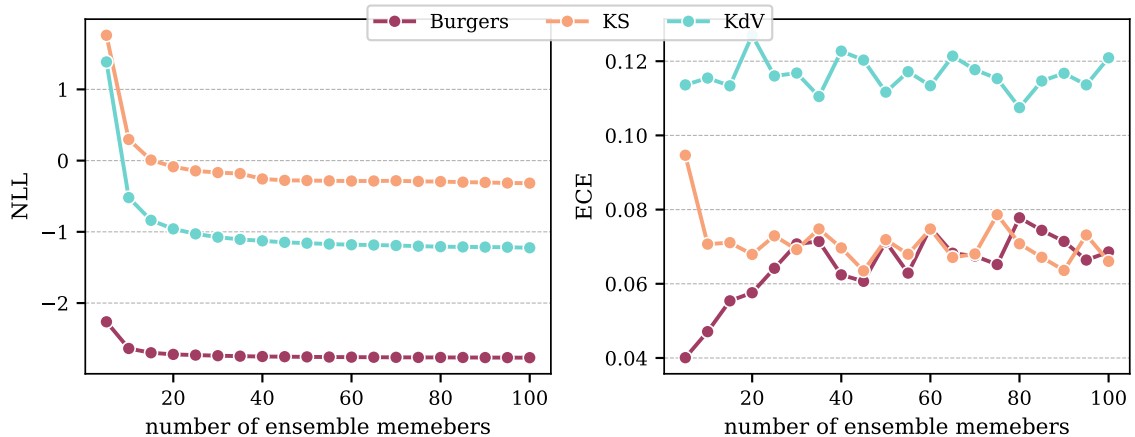

Figure 6: Caption

Table 9: RMSE for Burgers, KS and KdV PDEs for different Neural Solvers. The mean and std are computed for four different random weights initialization seeds. Solvers are unrolled for 320 steps.

| Model | RMSE (↓) | | |
| --- | --- | --- | --- |
| | *Burgers* | *KS* | *KdV* |
| BARNN | $\mathbf{0.019}^{\pm 0.004}$ | $0.217^{\pm 0.047}$ | $0.061^{\pm 0.026}$ |
| ARD | $0.020^{\pm 0.009}$ | $\mathbf{0.161}^{\pm 0.052}$ | $0.063^{\pm 0.030}$ |
| Dropout | $0.019^{\pm 0.008}$ | $0.172^{\pm 0.026}$ | $\mathbf{0.053}^{\pm 0.022}$ |
| Perturb | $0.022^{\pm 0.005}$ | $0.222^{\pm 0.067}$ | $0.090^{\pm 0.036}$ |
| Refiner | $0.042^{\pm 0.024}$ | $0.407^{\pm 0.402}$ | $0.775^{\pm 1.581}$ |

level. However, as discussed in Section 4.2, RMSE alone is insufficient for evaluating neural PDE solvers, particularly in uncertainty quantification scenarios, as low RMSE values can obscure inaccurate uncertainty estimates from the solver (see Figure 3).

**Test number of ensemble members in BARNN**  We extend the analysis presented in Section 4.2 by examining the convergence behaviour with varying numbers of ensemble members. Figure 6 illustrates how the negative log-likelihood (NLL) and expected calibration error (ECE) evolve as the ensemble size increases. Similar to the trend observed in RMSE (refer to Figure 4), the NLL and ECE metrics stabilize after approximately 30 ensemble members. This indicates that, beyond this point, adding more members yields diminishing returns in terms of performance improvement. These results highlight that BARNN is capable of generating precise and well-calibrated predictions, offering sharp uncertainty estimates without requiring a large ensemble. The saturation of these statistics reinforces the efficiency of BARNN, making it a robust method for uncertainty quantification with minimal computational overhead.

**Solution rollouts**  Figures 7, 8, and 9 showcase examples of 1-dimensional rollouts over 320 steps, generated using 100 ensemble members with BARNN. As anticipated, the error accumulates over time, which is a characteristic of the autoregressive nature of the PDE solver. However, the model's variance also increases in tandem, highlighting its ability to identify regions with higher errors and demonstrating impressive adaptability. Despite the growing errors, the overall solution remains closely aligned with the predictive mean, reflecting strong pointwise accuracy. Meanwhile, the variance effectively captures the rise in errors, providing a reliable indication of uncertainty and reinforcing the model's adaptability in challenging regions of the solution space. This dual capability of maintaining accuracy while adapting to uncertainty underscores the robustness of the BARNN approach.

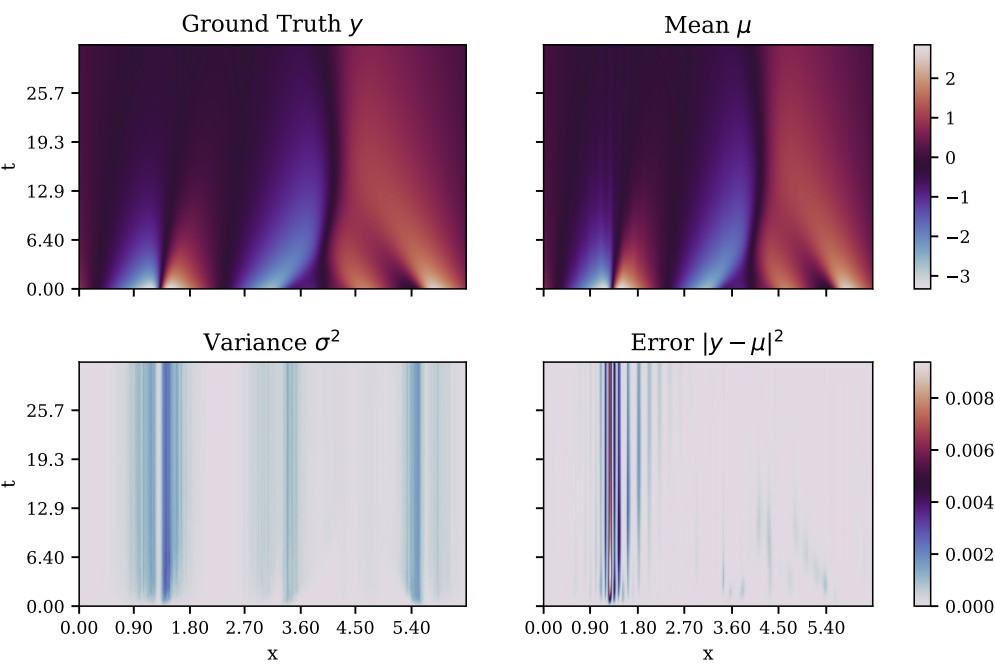

Figure 7: Exemplary of 1-dimensional Burgers rollout.

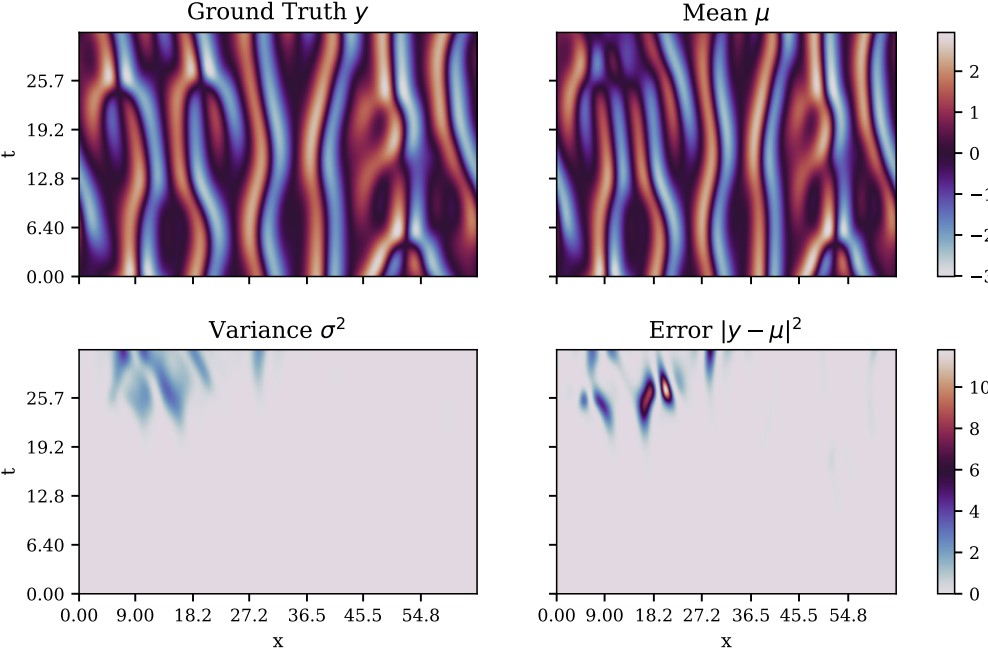

Figure 8: Exemplary of 1-dimensional Kuramoto-Sivashinsky rollout.

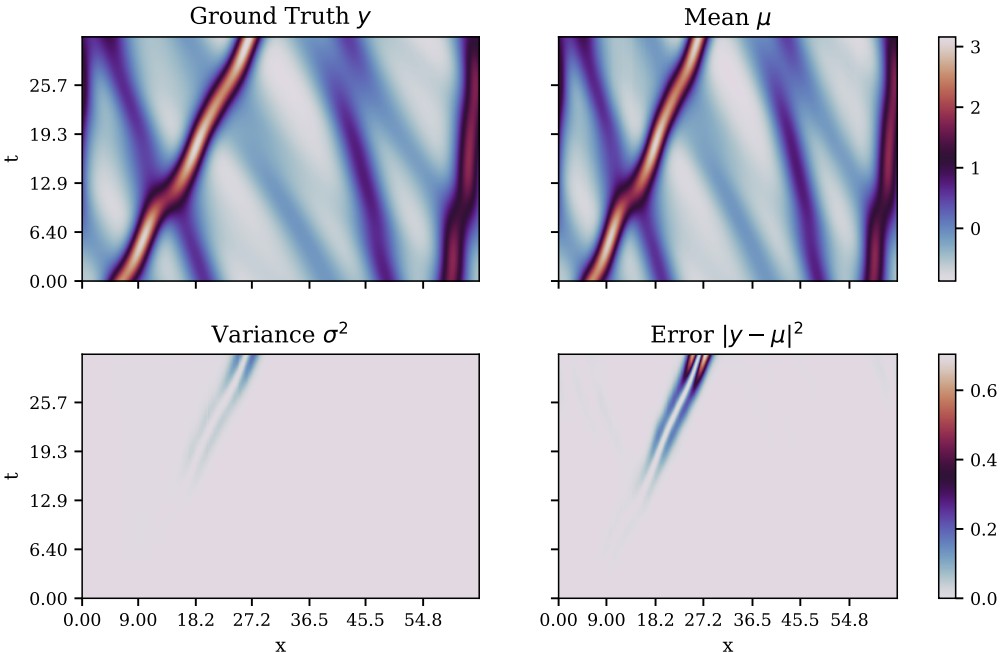

Figure 9: Exemplary of 1-dimensional Korteweg de Vries rollout.

## C. BARNN Molecules Application

This section shows how BARNN can be used to build Bayesian Recurrent Neural Networks, provides all the details for reproducing the experiments, and shows additional results.

### C.1. Data Generation and Metrics

**Data Generation:** The experiments focus on unconditional molecule generation using the SMILES syntax, a notation used to represent the structure of a molecule as a line of text. We use the data set provided in (Özçelik et al., 2024), which contains a collection of 1.9 M SMILES extracted from the ChEMBL data set (Zdrazil et al., 2024). Before training, the SMILES strings were tokenized using a regular expression, containing all elements and special SMILES characters, e.g., numbers, brackets, and more. Therefore, each atom, or special SMILES character, is represented by a single token.

**Metrics:** To evaluate the performance of the language models, we focus on different metrics:

1. **Validity**: the percentage of generated SMILES corresponding to chemically valid molecules

2. **Diversity**: the percentage of unique Murcko scaffolds among generated molecules.

3. **Novelty**: the percentage of generated molecules absent from the training dataset

4. **Uniqueness**: the percentage of structurally-unique molecules among the generated designs

Moreover, we compute the Wasserstein distance between the molecular properties distribution for the language model and the ChEMBL dataset to ensure molecular properties are matched. Similarly to (Segler et al., 2018) we analyse the molecular weight, H-bond donors and acceptors, rotatable bonds, LogP, and TPSA.

### C.2. Model Architectures, Hyperparameters, and Computational Costs

All models were trained for 12 epochs using Adam with $2 \cdot 10^{-4}$ of learning rate and $10^{-8}$ of weight decay for regularization. The training was distributed across four Tesla P100 GPUs with 16-GB of memory each, and we used a batch size of 256 for memory requirements. Approximately 8 hours are needed to train one model.

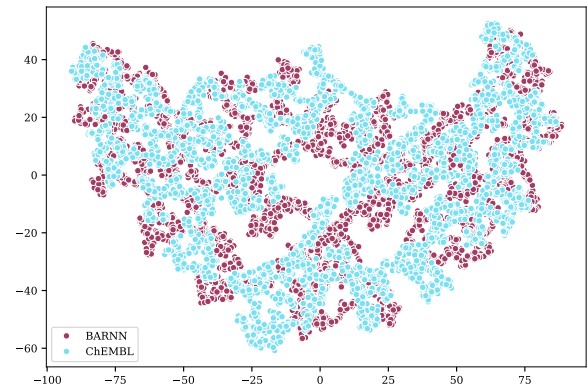

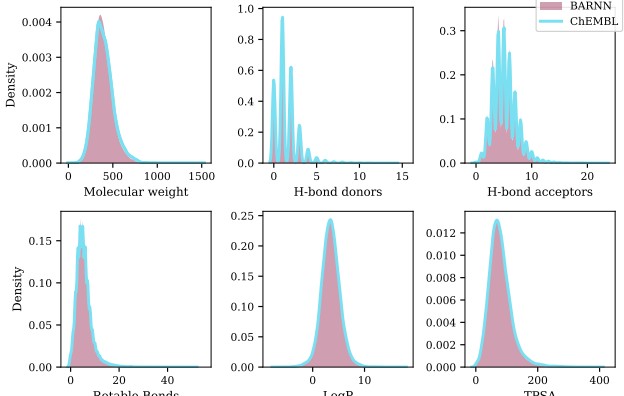

Figure 10: t-SNE representation of different molecular property descriptors for BARNN and chEMBL molecules. The distributions of both sets overlap.

Figure 11: Distribution density for each molecular property descriptor for BARNN and chEMBL molecules. The distributions for different descriptors overlap significantly.

We perform all experiments with an LSTM gate mechanism (Schmidhuber et al., 1997) for recurrent networks. The tokens were mapped to a one-hot encoding of the size of the vocabulary, processed by the language model, and mapped back to the vocabulary size by a linear layer. We used three LSTM-type layers with the hidden dimension of 1024 accounting for 25 M parameters model overall. For the dropout LSTM we used 0.2 as dropout coefficient, following indications from previous works (Segler et al., 2018). For BARNN the encoder is composed of two fully connected (linear) layers of size 128 interleaved by LeakyReLU activation which we found to be the best, and a final sigmoid activation to return the dropout probabilities $p_t$. The dropout rates can be obtained by computing $\alpha_t = p_t/(1 - p_t)$ elementwise. Once models were trained, $50K$ molecules were sampled to compute the statistics reported in the experiment section.

## C.3. Additional Results

In this section, we provide additional results for molecule generation using BARNN.

**Molecular properties predictions**   We compute six molecular property descriptors—namely Molecular Weight (MW), H-bond donors (HBDs), H-bond acceptors (HBAs), Rotable Bonds, LogP, and Topological Polar Surface Area (TPSA)—for both BARNN-generated molecules and the chEMBL dataset. To evaluate whether the molecular properties of the BARNN-generated molecules align with those of the chEMBL data, we employ t-SNE (t-Distributed Stochastic Neighbor Embedding) (Van der Maaten & Hinton, 2008), a dimensionality reduction technique, to visualize the two-dimensional latent space (Figure 10). The t-SNE plot reveals that the distributions of the two sets significantly overlap, suggesting that BARNN effectively captures the underlying molecular properties. For a more detailed analysis, we perform Kernel Density Estimation (KDE) for each molecular property descriptor and present the resulting distributions for both BARNN and chEMBL in Figure 11. The KDEs from both datasets almost perfectly overlap, even for distributions with multiple peaks (e.g., H-bond donors and acceptors), indicating that BARNN not only captures the overall distribution but also the single characteristics of the molecular properties with high fidelity.

**Robustness to temperature change**   Temperature is a crucial hyperparameter in recurrent neural networks that influences the randomness of predictions by scaling the logits before applying the softmax function. As $T \to 0$, the model tends to select the most likely element according to the Conditional Language Model (CLM) prediction, leading to more deterministic outputs. Conversely, as the temperature increases, the selection becomes more random, as higher temperatures flatten the logits. In our analysis, we evaluate the robustness of BARNN to temperature variations during the sampling process, specifically examining the validity and uniqueness of the generated sequences, as shown in Figure 13. Across all models, we observe a general trend: both validity and uniqueness decline as the temperature increases. At lower temperatures, BARNN outperforms the baseline models, maintaining higher validity and uniqueness. However, at very high temperatures, BARNN and the base LSTM show comparable performance, with both models achieving similar levels of validity and uniqueness.

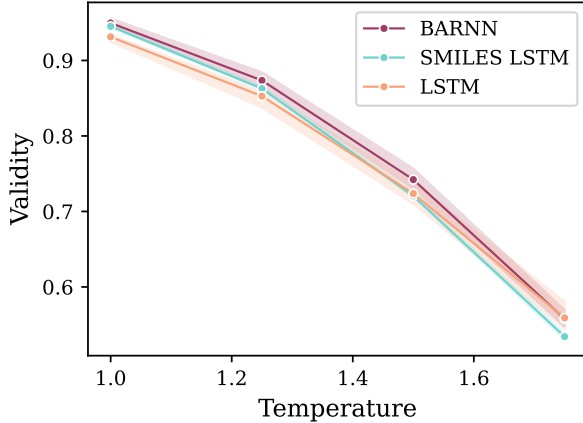
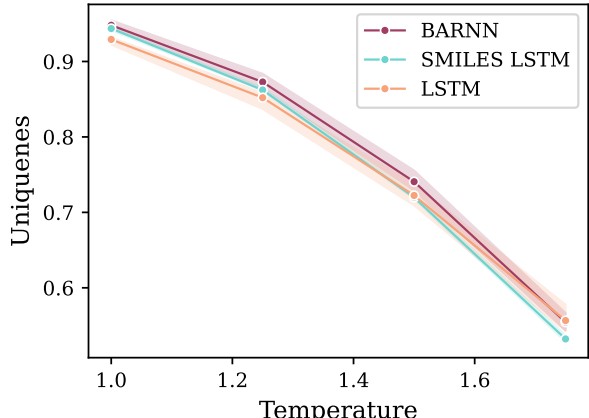

Figure 12: t-SNE representation of different molecular property descriptors for BARNN and chEMBL molecules. The distributions of both sets overlap.

Figure 13: Distribution density for each molecular property descriptor for BARNN and chEMBL molecules. The distributions for different descriptors overlap significantly.

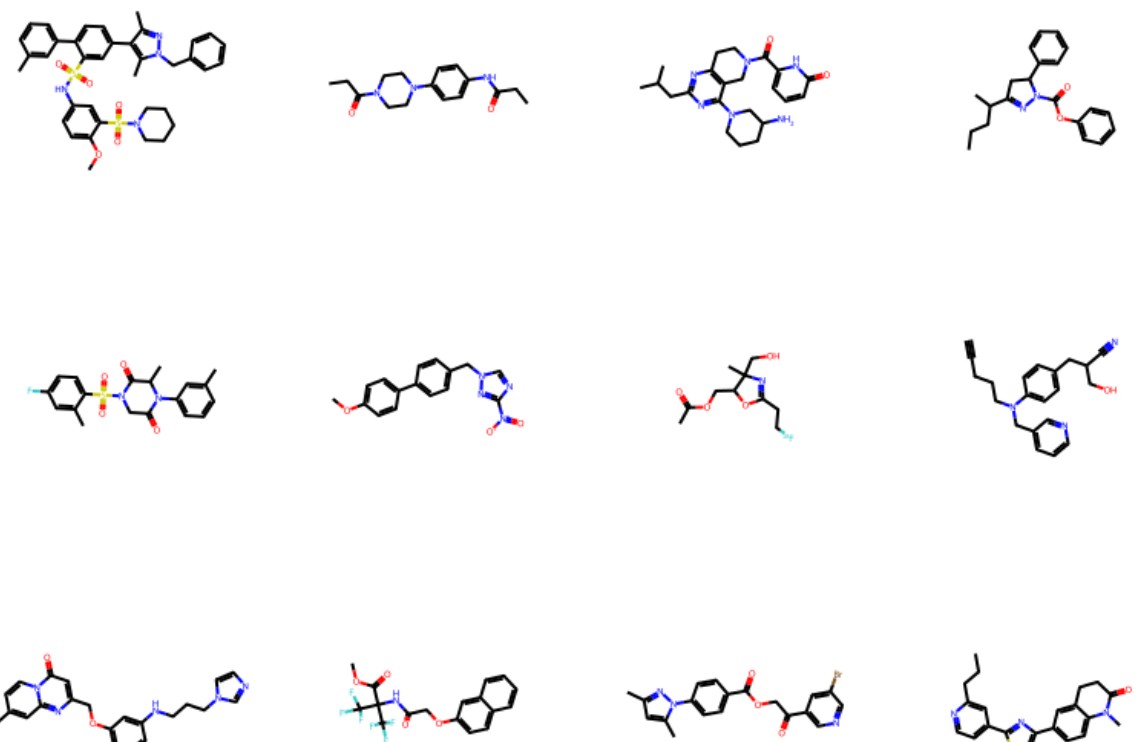

Figure 14: Samples of generated molecules with BARNN.

# D. Algorithms and Software

This section reports the pseudocode algorithms for training and performing inference using a BARNN model and the software used.

### D.1. Pseudocodes

**Algorithm 1** *Local reparametrization trick*: Given $\boldsymbol{H}$ minibatch of activations for layer $l = 1, \ldots, L$ use the local reparametrization trick to compute the linear layer output. It requires the dropout rate coefficient $\alpha^l$ for layer $l$, and the Neural Operator static weights $\boldsymbol{\Omega}_l$ for layer $l$. For the first layer, activation corresponds to the state input.

$\boldsymbol{M} \leftarrow \alpha^l \boldsymbol{H} \, \boldsymbol{\Omega}_l$ {Compute the mean}
$\boldsymbol{V} \leftarrow (\alpha^l \boldsymbol{H})^2 \, \boldsymbol{\Omega}_l^2$ {Compute the variance, power is taken elementwise}
$\boldsymbol{E} \sim \mathcal{N}(\boldsymbol{0}, \mathbb{I})$ {Sample a Gaussian state}
**return** $\boldsymbol{M} + \sqrt{\boldsymbol{V}} \odot \boldsymbol{E}$

---

**Algorithm 2** *Training Bayesian Neural PDE Solvers*: For a given batched input data trajectory, Neural Solver and variational posterior encoder, we draw a random timepoint t, get our input data trajectory, compute the dropout rates with the posterior encoder, perform a forward pass using the local-reparametrization trick, and finally, perform the supervised learning task with the according labels plus the KL regularization. $\{\boldsymbol{y}_0^k, \ldots, \boldsymbol{y}_T^k\}_{k=1}^N$ are the data trajectories, NO is the neural operator architecture with $\boldsymbol{\Omega}$ static weights, and $E_\psi$ is the variational posterior encoder.

**while** not converged **do**
    $t \leftarrow \mathcal{U}[1, T]$ {Draw uniformly a random ending point}
    $\boldsymbol{\alpha}_t \leftarrow E_\psi(\boldsymbol{y}_{t-1}^k)$ {Compute dropout rates for each NO layer}
    $\hat{\boldsymbol{y}}_t^k \leftarrow \text{NO}(\boldsymbol{y}_{t-1}^k; \boldsymbol{\Omega}, \boldsymbol{\alpha}_t)$ {Apply local reparametrization trick to linear layers}
    $\mathcal{L}(\boldsymbol{\Omega}, \boldsymbol{\psi}) \leftarrow \|\hat{\boldsymbol{y}}_t^k - \boldsymbol{y}_t\|^2 - D_{KL}(\boldsymbol{\alpha}_t)$ {Compute the loss using $D_{KL}$ from eq. (9)}
    Optimize $\boldsymbol{\Omega}, \boldsymbol{\psi}$ by descending the gradient of $\mathcal{L}$
**end while**

---

**Algorithm 3** *Training Bayesian Recurrent Neural Networks*: For a given batched input and output data taken from the sequence we compute the dropout rates with the posterior encoder, perform a forward pass using the local-reparametrization trick, and finally, perform the supervised learning task with the according labels plus the KL regularization. $\{\boldsymbol{y}_0^k, \ldots, \boldsymbol{y}_T^k\}_{k=1}^N$ are the sequences tokenized, RNN is the recurrent architecture with $\boldsymbol{\Omega}$ static weights, $\boldsymbol{o}_t$ is the RNN probability output at step $t$, and $E_\psi$ is the variational posterior encoder.

**while** not converged **do**
    $\boldsymbol{h} \leftarrow 0$ {Initialize to zero hidden state}
    **for** $t \in [0, \ldots, T-1]$ **do**
        $\boldsymbol{\alpha}_t^y \leftarrow E_\psi(\boldsymbol{y}_t)$ {Compute dropout rates for each RNN layer}
        $\boldsymbol{\alpha}_t^h \leftarrow E_\psi(\boldsymbol{h})$ {Compute dropout rates for each RNN layer}
        $\boldsymbol{o}_{t+1}, \boldsymbol{h} \leftarrow \text{RNN}(\boldsymbol{y}_t, \boldsymbol{h}; \boldsymbol{\Omega}, \boldsymbol{\alpha}_t^y, \boldsymbol{\alpha}_t^h)$ {Apply RNN forward with local reparametrization trick}
    **end for**
    $\mathcal{L}(\boldsymbol{\Omega}, \boldsymbol{\psi}) \leftarrow \text{CrossEntropyLoss}(\boldsymbol{o}_{1:T}, \boldsymbol{y}_{1:T}) - D_{KL}(\boldsymbol{\alpha}_t)$ {Compute the loss}
    Optimize $\boldsymbol{\Omega}, \boldsymbol{\psi}$ by descending the gradient of $\mathcal{L}$
**end while**

---

### D.2. Software

We perform the PDEs experiment using PINA (Coscia et al., 2023) software, which is a Python library based on PyTorch (Paszke et al., 2019) and PyTorch Lightning (Falcon & The PyTorch Lightning team, 2019) used for Scientific Machine Learning and includes Neural PDE Solvers, Physics Informed Networks and more. For the Molecules experiments we used PyTorch Lightning (Falcon & The PyTorch Lightning team, 2019) and RDKit for postprocessing analysis of the molecules.

