# OpenReview forum: "BARNN: A Bayesian Autoregressive and Recurrent Neural Network"
_ICML.cc/2025/Conference — ICML 2025 poster_

### Official Review · Reviewer_WqtE · 2025-03-08

**Overall Recommendation:** 3

**Summary:**

This paper introduces a Bayesian version of an RNN by modeling the time-dependent weights as a reparameterized random variable, where the posterior is learned in an amortized way using a time-dependent ELBO. A specific variational posterior parameterization is provided, where each layer’s weights are parameterized via multiplicative Gaussian dropout with a time-dependent scaling factor as parameters. These dropout scaling factors evolve overtime and are generated by an encoder network that conditions on past observations. Instead of a fixed isotropic Gaussian prior, the model employs a Temporal Variational Mixture of Posteriors (tVAMP) prior, which aggregates past variational posteriors to improve uncertainty modeling and calibration. The framework is applied to Neural PDE solvers and molecular language models (unconditional SMILES generation), demonstrating better uncertainty quantification (lower NLL) in PDE modeling and higher validity in unconditional molecular generation.

**Claims And Evidence:**

Overall, all the claim are supported by clear and convincing evidence.


Strong claim:
- BARNN offers greater accuracy -> Table 1
- BARNN calibrated and sharp uncertainty estimates -> Table 1, Fig. 2
- BARNN excels in modelling long-range molecular dependencies compared to related methods. -> Fig. 5
- BARNN is the first approach that transforms any autoregressive or recurrent model into its Bayesian version with minimal modifications. -> see related work below

Weak claim(promise):
- autoregressive models are also prone to overfitting to the specific tasks they are trained on, challenging their application outside their training domain. -> Not explicitly demonstrated

**Essential References Not Discussed:**

I think literature [1] , which also introduces a variational Bayesian scheme for RNNs, is highly relevant and should be discussed.

[1] https://arxiv.org/pdf/1704.02798

**Experimental Designs Or Analyses:**

- Neural PDE solver benchmark:

    - The use of autoregressive models for solving time-dependent PDEs is well-motivated, as these models naturally capture temporal dependencies, and evaluating uncertainty estimation is important.
    - The uncertainty calibration metrics (NLL/ECE) are appropriate and commonly used in probabilistic modeling.

- SMILES molecular generation benchmark:
    - Autoregressive models are widely used in molecular generation, making SMILES-based generation a standard benchmark.
Bayesian uncertainty is particularly valuable for molecular design and exploration, where uncertainty-aware sampling improves model reliability.
    - The Wasserstein distance, t-SNE visualization (Fig. 10), and property histograms (Fig. 11) faithfully capture different aspects of distribution similarity—global shape (Wasserstein), structural overlap (t-SNE), and individual property distributions (histograms).

**Methods And Evaluation Criteria:**

The proposed methods and evaluation criteria make sense to me, for detailed elaboration please see below "Experimental Designs Or Analyses".

**Other Comments Or Suggestions:**

-  Eq.~8: should it be $E_{\phi}(\boldsymbol y_{0:t-1}^k)$ (i.e. a $E_{\phi}(\cdot)$ is missing)?

**Other Strengths And Weaknesses:**

Strength
- **A unified framework for Bayesian RNNs and autoregressive models**
The paper provides a complete pipeline, making it potentially applicable to various autoregressive tasks.

Weaknesses
- **Hard to position in context**
    - The incorporation of Bayesian methods into RNNs has a long history, with key prior work (see "Essential References Not Discussed") that is not acknowledged or discussed.
    - Combined with the presentation style, this gives the impression of overclaiming novelty, as if applying Bayesian inference to RNNs is itself a novel contribution.
- **Ablation study of the prior**:  The tVAMP prior is a key contribution, yet the paper does not include an ablation study comparing it to a simpler isotropic Gaussian prior.
- **Motivation for using RNNs over stronger sequence models**: While this may not be a direct weakness, it is unclear why the paper focuses on Bayesian RNNs when stronger sequence models (e.g., S4) exist.

**Questions For Authors:**

- **Intuition Behind Long-Term Dependency Claims**
    - BARNN is claimed to help capture long-term dependencies. In the SMILES generation task, do the authors have any intuition on why Bayesian modeling specifically mitigates the ring-closing issue compared to deterministic baselines?
- **Practical Setting of
$N$ in Eq.~(8)**:
    - is the $N$ in Eq.\~(8) pragmatically set by Batch size of training data or the whole data size?

**Relation To Broader Scientific Literature:**

This work is related to Bayesian deep learning and Bayesian autoregressive models. The proposed Bayesian RNN structure, which uses a multiplicative scaling factor scheme, provides a generalizable framework for transforming autoregressive models and RNNs into their Bayesian counterparts. This builds on prior work in variational Bayesian methods [1] and VAMP priors [2], extending them to time-dependent settings. The method is of pragmatic interest due to its broad applicability in uncertainty-aware sequence modeling.


[1] https://proceedings.neurips.cc/paper/2015/hash/bc7316929fe1545bf0b98d114ee3ecb8-Abstract.html
[2] https://arxiv.org/abs/1705.07120

**Theoretical Claims:**

This paper does not have any theoretical claim.

The ELBO derivation and the marginalized prior are plausible to me.

---

> ### Author Rebuttal · Authors · 2025-03-31
>
> We appreciate the reviewer for their time and valuable feedback. In particular, we are delighted that the reviewer believes our work *is of pragmatic interest due to its broad applicability in uncertainty-aware sequence modeling*, and that their appreciate the experiment section. We answer the questions raised below point by point:
>
> **Weak claim(promise):
> autoregressive models are also prone to overfitting to the specific tasks they are trained on, challenging their application outside their training domain. -> Not explicitly demonstrated**
> - We thank the reviewer for analysing in depth our results section, we appreciate the time and effort. We agree that we did not stress in our experiment analysis autoregressive model overfitting. However, we want to highlight that our Table 2 results show the overfitting behaviour of autoregressive models. In particular, SMILES LSTM (it has dropout on the recurrent layers) and BARNN can be thought of as regularized versions of the vanilla LSTM. The latter, compared to the regularised models, is more prone to reproduce molecules already seen during training, shown by lower novelty and uniqueness, which is an indication of overfitting to the training data distribution.
>
> **Essential References Not Discussed:
> I think literature [1] , which also introduces a variational Bayesian scheme for RNNs, is highly relevant and should be discussed.**
> - We thank the reviewer for pointing out the methodology to us, which shares the similarity of modelling uncertainties in RNNs, and we will add in section 2. However, differently from our methodology, the approach is only suited to RNNs, while our approach can be applied to various autoregressive or recurrent models. Furthermore, but most importantly, the model in [1] can hardly scale to large networks mainly due to two reasons (i) weights are sampled directly, in contrast we use the local-reparametrization allowing scaling to large networks (ii) the number of parameters doubles because they model mean and standard deviation directly, while we use a variance proportional to the mean.
>
> **Weaknesses:
> Hard to position in context: The incorporation of Bayesian methods into RNNs has a long history, with key prior work (see "Essential References Not Discussed") that is not acknowledged or discussed.**
> - See above in “Essential References Not Discussed” answer.
>
> **Weaknesses: Combined with the presentation style, this gives the impression of overclaiming novelty, as if applying Bayesian inference to RNNs is itself a novel contribution.**
> - We thank you for the comment, and we do not want to claim that applying Bayesian inference to RNNs is our novel contribution. However, to the best of our knowledge, we are the first approach that can transform any autoregressive or recurrent model into a Bayesian one with minimal modification, obtaining similar or superior performances to non-Bayesian counterparts, scale to large networks and at the same time provide UQ.
>
> **Weaknesses: Ablation study of the prior: The tVAMP prior is a key contribution, yet the paper does not include an ablation study comparing it to a simpler isotropic Gaussian prior.**
> - We thank the reviewer for the suggestion, which is shared with reviewer 4bFX. We conducted a test on a synthetic time-series dataset to show how tVAMP excels over standard log uniform prior and fixed dropout rates. A simple isotropic Gaussian would not lead to a KL which is independent on the likelihood model parameters, therefore losing the regularization property of VD. Please see in reviwer's 4bFX answer the result table and explanation (not inserted here for words limit).
>
> **Weaknesses: Motivation for using RNNs over stronger sequence models: While this may not be a direct weakness, it is unclear why the paper focuses on Bayesian RNNs when stronger sequence models (e.g., S4) exist.**
> - We thank the reviewer for the comment, also suggesting a possible application to BARNN for SSMs. We wanted to focus on RNNs because the model in eq. (2) (3) is heavily recurrent. Of course, extending to SSMs and Transformers models would be interesting, but a specific ancoder $E$ is needed, and further research needs to be conducted. In summary, with the paper, we wanted to show a general methodology for Bayesian sequence modelling, hoping to inspire potential applications to future work.
>
> **Eq.~8: should it be $E_{\phi}(\mathbf{y}_{0:t-1}^k)$?**
> - Yes, we write it more compactly as $\alpha^l_t(\mathbf{y}_{0:t-1}^k)$, we will add a note on the paper, thank you for spotting it!
>
> **Intuition Behind Long-Term Dependency Claims**
> - We thank you for the interesting question, and we claim that this is due to the time-adaptivity of the model weights. Indeed, both SMILES LSTM (with Dropout) and standard LSTM obtain similar performances, worse than BARNN, which only has temporal adaptivity differently from them.
>
> **Practical Setting of $N$ in Eq.~(8)**
> - Correct! $N$ is set to the batch size. We will update the paper to be clearer.

---

### Official Review · Reviewer_fti8 · 2025-03-11

**Overall Recommendation:** 3

**Summary:**

This paper addresses the problem of uncertainty quantification with autoregressive and recurrent neural neworks. Variational Bayes method is applied to infer the posterior distribution of network parameters, and further techniques are developed using variational dropout methods. Applications are made to PDE solving and molecular generation.

**Claims And Evidence:**

The motivation is clear and the use of variational inference for infering autoregressive and recurrent neural network parameters is a reasonable choice.

Eq.(3) on the factorization form of variational posterior appears to a good choice, which also associates with the idea of modular Bayes. More to be discussed later.

For Eq.(6), why is the normal distribution assumed to have variance being the square of its mean?

**Essential References Not Discussed:**

Here is an example reference on modular Bayes:

Bayarri, M. J., J. O. Berger, and F. Liu. "Modularization in Bayesian analysis, with emphasis on analysis of computer models." Bayesian Anal. 4(1): 119-150

**Experimental Designs Or Analyses:**

The experimental designs look nice and the choices of applications are real-world driven.

I do have a question regarding he confidence intervals presented in Figure 2. It is said in the paper that the presented CIs are 99.7%, which are still very narrowly concentrating near the mean prediction/posterior predictive mean. To me, it appears that the variational inference approach might have signficantly underestimated the uncertainty.

**Methods And Evaluation Criteria:**

Evaluations are reasonable, ranging over both metrics on predicitve accuracy and metrics on uncertainty quantification.

**Other Comments Or Suggestions:**

N/A.

**Other Strengths And Weaknesses:**

See above.

**Questions For Authors:**

See Claims And Evidence & Experimental Designs Or Analyses.

**Relation To Broader Scientific Literature:**

The design of the factorization format of variational posterior in Eq.(3) reminds of the line of work on modular Bayes inference, where we deliberately cut off certain dependencies between parameter blocks and parts of observations, in order to improve computational efficiency / reduce potential influence of model misspecification due to incorrect inclusion of certain dependencies.

**Theoretical Claims:**

The derivations of ELBO and related formulas look correct.

---

> ### Author Rebuttal · Authors · 2025-03-31
>
> We appreciate the reviewer's time and effort in making constructive comments and providing us with additional references. We are glad that the reviewer finds our experiments *nice* and acknowledges that *the choices of applications are real-world driven*. We answer below the questions raised by the reviewer, and we also ask if there are other concerns about the paper which not lead to a higher score in the initial review round; in that case, we will be glad to discuss.
>
> **For Eq.(6), why is the normal distribution assumed to have variance being the square of its mean?**
> - Thank you for pointing out this crucial step in our methodology. We choose to model the variance as the square of the mean, as also done in vanilla Variational Dropout, for efficiency proposes. This parametrization allows us to not use extra model parameters, permitting scaling to very large neural networks. Also, this approach allows us not to change the original (non-Bayesian) model and add the BARNN methodology on top of it, which, as  4bFX highlighted, is “easy to implement”
>
> **I do have a question regarding he confidence intervals presented in Figure 2. It is said in the paper that the presented CIs are 99.7%, which are still very narrowly concentrating near the mean prediction/posterior predictive mean. To me, it appears that the variational inference approach might have signficantly underestimated the uncertainty.**
> - We thank the reviewer for the comment on the CIs. We agree with the reviewer that in the picture, the confidence interval seems concentrated near the mean prediction/posterior predictive mean. However, this is mainly a plotting issue because we used a very thick line width to have three nice displayed figures side by side. To convince you, please look at the ECE and NLL in Table 1. Underestimating the variance (as in the case of Refiner, for example) would lead to a very high ECE and exploding NLL (which is not our case).
>
> **The design of the factorization format of variational posterior in Eq.(3) reminds of the line of work on modular Bayes inference, where we deliberately cut off certain dependencies between parameter blocks and parts of observations, in order to improve computational efficiency / reduce potential influence of model misspecification due to incorrect inclusion of certain dependencies.**
> - We thank the reviewer for pointing out this reference, which we believe is relevant. Indeed, we cut off non-causal dependencies in eq (3) (i.e. the current weights are only dependent on the past). We will look more in depth into modular bayes inference and add a connection to eq. (3) in the final manuscript. If there are other concerns or suggestion, we are happy to discuss.

---

### Official Review · Reviewer_SBfz · 2025-03-12

**Overall Recommendation:** 2

**Summary:**

This article proposes a new Bayesian recurrent neural network, mainly by introducing variational dropout into recurrent neural networks. Experiments can prove the model's ability to quantify uncertainty.

**Claims And Evidence:**

This paper builds a variational Bayesian autoregressive and recurrent neural network. BARNNs aim to provide a principled way to turn any autoregressive or recurrent model into its Bayesian version. BARNN is based on the variational dropout method, allowing us to apply it to large recurrent neural networks as well.  I think is correct .

**Essential References Not Discussed:**

N/A

**Experimental Designs Or Analyses:**

The experiments can verify the method is effective and are taken on different tasks.

**Methods And Evaluation Criteria:**

Quantifying uncertainty in autoregressive models is a crucial and key issue.

**Other Comments Or Suggestions:**

N/A

**Other Strengths And Weaknesses:**

Strongth
1. Uncertainty modeling is a key problem in neural networks, and I think the paper as a whole is workable, and the results look good.

2. The experiments are sufficient to demonstrate the validity of the model.

3. The paper is well wrightten.

Weakness
1. The proposed method can be seen as an application of variational dropout to an autoregressive model but does not provide any new insight.

2. Autoregressive models based on attention mechanisms are not discussed.

**Questions For Authors:**

1. Is there any difference between probabilistic modeling of autoregressive models and normal neural network models, like convolutional neural networks?

2. Why not use Transformer, a more general and powerful autoagressive neural network?

3. Why not experiment on text to better illustrate the effectiveness of the model on large data sets?

**Relation To Broader Scientific Literature:**

Some work attempts to probabilistic model attention in order to obtain good uncertainty estimation ability.

[1] Bayesian Attention Modules; Xinjie Fan, Shujian Zhang, Bo Chen, and Mingyuan Zhou; NeurIPS 2020: Advances in Neural Information Processing Systems, Dec. 2020.
[2] Deng, Yuntian, Yoon Kim, Justin Chiu, Demi Guo, and Alexander Rush. "Latent alignment and variational attention." Advances in neural information processing systems 31 (2018).

**Theoretical Claims:**

A posteriori inference technique based on variational autoencoder, I think, is correct

---

> ### Author Rebuttal · Authors · 2025-03-31
>
> We thank the reviewer for their time and comments, and we appreciate that the reviewer finds the paper *well written* with the experiments *sufficient to demonstrate the validity of the model*. We address the reviewer's comments point by point below in the hope of clarifying some misunderstandings, improving the paper and eventually raise their score:
>
> **Some work attempts to probabilistic model attention in order to obtain good uncertainty estimation ability**
> - We thank the reviewer for the references; we find  Bayesian Attention Modules very relevant and will cite them in the related works. A key difference between Bayesian Attention Modules and our methodology is that our method explicitly models uncertainties in time using a joint state-weight model and applies to any autoregressive or recurrent model, while Bayesian Attention Modules are a technology specifically tailored to Transformers models, and the Bayesian weights are static.
>
> **Weakness**
> **The proposed method can be seen as an application of variational dropout to an autoregressive model but does not provide any new insight.**
> - We thank the reviewer for their time in reading the paper and finding parallelisms with Variational Dropout (VD). We understand that our method shares similarities with VD, from which we inherit the scalability to large networks. However, in variational dropout methods (i) the weights do not evolve in time, and (ii) the prior is empirical, while we provide a time-dependent new prior, which is the best for the objective in eq. (4). As reviewer 4bFX highlighted, *the derived prior (tVAMP) and time-dependent dropout mechanics feel both novel and easy to implement*, and the reviewer WqtE believes that our methodology *is of pragmatic interest due to its broad applicability in uncertainty-aware sequence modeling*. Finally, we also found that static weights lead to less accurate uncertainty metrics and adaptivity. For example, see Table 1, where ARD Dropout or Dropout shows lower accuracy in terms of NLL and ECE compared to our methodology.
>
> **Autoregressive models based on attention mechanisms are not discussed.**
> - We thank the reviewer for pointing out attention-based models, which we agree we did not mention exhaustively. We tried to cover a wide variety of references on autoregressive models, not focusing only on a specific model. Nevertheless, we did mention the Vaswani, A. et al. Attention Is All You Need paper for transformer reference, and the Radford, A., et al. Language Models are Unsupervised Multitask Learners for moder decoder-only language models. If the reviewer thinks we missed other relevant papers which will improve the manuscript, we will be happy to add them if pointed to.
>
> **Questions For Authors:**
> **Is there any difference between probabilistic modeling of autoregressive models and normal neural network models, like convolutional neural networks?**
> - We thank the reviewer for the very interesting question. We did find that standard techniques used to model uncertainties in non-autoregressive models tend to obtain less accurate uncertainty metrics when applied to autoregressive solvers (see, for instance, Table 1. where Ensemble Dropout does not obtain as good performances as our methodology). This is mainly due to error accumulation and long-term dependencies, which non-autoregressive models do not need to deal with.
>
> **Why not use Transformer, a more general and powerful autoaggressive neural network?**
> - We thank the reviewer for the question, which we are glad to discuss. In our experiment, we focused in the AI4Science field since we believe uncertainty quantification will be key there. In the AI4Science field, we choose PDEs and Molecule generation as tasks since they are very different and use different models. For PDEs, autoregressive models are mostly based on multiple-step predictions using a Neural Operator. For molecule generation, RNNs were first applied, and just more recently, Transformers have been used, and it is not clear that transformers are more powerful in this specific task. In summary, with the paper, we wanted to show a general methodology for Bayesian sequence modelling,  hoping to inspire potential applications to future work given the simplicity to implement and the generality of the methodology.
>
> **Why not experiment on text to better illustrate the effectiveness of the model on large data sets?**
> - We agree with the reviewer that the NLP task is an interesting avenue of research. Even though not an NLP task, we did employ text for the molecule generation task where the SMILES syntax is learned by the model to generate new molecules. For a more specific NLP task (e.g. language translation), a complication for us was how to define uncertainty properly, which seems to be an ongoing open research. Indeed, we choose our experiment tailored to support each claim in the paper, as reviewer WqtE also highlighted (*all the claim are supported by clear and convincing evidence*).

---

> > ### Comment · Reviewer_SBfz · 2025-04-02
> >
> > Thanks for your reply.
> >
> > What is unique about your autoregressive models for time series uncertainty modeling,
> >
> >
> > [1] Desai, A., Freeman, C., Wang, Z., and Beaver, I. Timevae:
> > A variational auto-encoder for multivariate time series
> > generation. arXiv preprint arXiv:2111.08095, 2021.
> >
> > [2] Kollovieh, M., Ansari, A. F., Bohlke-Schneider, M.,
> > Zschiegner, J., Wang, H., and Wang, Y. Predict, refine,
> > synthesize: Self-guiding diffusion models for probabilistic time series forecasting. In NeurIPS, 2023.
> >
> >
> > [3] Probabilistic Transformer For Time Series Analysis, NrurIPS 2021

---

> > > ### Author Response · Authors · 2025-04-06
> > >
> > > We thank the reviewer for reading our rebuttal and providing the interesting question. Regarding the question posed, we are, to the best of our knowledge, the first approach that can transform any autoregressive or recurrent model into a Bayesian one with minimal modification, obtaining similar or superior performances to non-Bayesian counterparts, scale to large networks and at the same time provide UQ.
> > >
> > > For time series (but in general for sequential data) this means that we can use any (not Bayesian) model (as the suggested references) and seamlessly integrate our Bayesian approach on top of it, without performance loss (actually improving in performance for what we observed in our experiments) and provide calibrated uncertainties, which is key for real-life applications.
> > > We also observed that the joint Bayesian model is capable to capture longer-time dependencies compared to the non-Bayesian counterpart, due to the adaptive in time Bayesian weights, as evidenced in the molecular experiment section.

---

### Official Review · Reviewer_4bFX · 2025-03-13

**Overall Recommendation:** 4

**Summary:**

This paper introduces BARNN, a framework designed to turn any autoregressive or recurrent deep learning model into a Bayesian version with minimal modifications. The authors propose jointly modeling both the states (e.g., tokens, PDE solutions, molecule strings) and the model’s weights as they evolve in time. By extending existing approaches to variational Bayesian inference, this work derives a novel temporal version of a variational lower bound and leverages a time-dependent extension of Variational Dropout. The work demonstrates BARNN in two main applications: (1) Neural PDE solvers for uncertainty quantification, and (2) RNN-based molecule generation, with experiments showing stronger calibration of uncertainties, improved long-range dependency handling, and better alignment with observed statistical properties of the underlying data.

**Claims And Evidence:**

1. BARNN can be applied to “any” autoregressive or recurrent model with minimal architectural changes. The paper shows that only a small set of modifications is needed: introducing time-dependent “dropout coefficients” and a new prior derived from the aggregated posterior. They illustrate this with PDE models and an LSTM-based molecule generator, both using standard training objectives (MSE for PDE forecasting, cross-entropy for next-token prediction).

2. BARNN provides better-calibrated and sharper uncertainties than existing methods. In experiments with PDE solvers, BARNN outperforms Monte Carlo Dropout, Input Perturbation, and other baselines on negative log-likelihood and expected calibration error.

3. BARNN enhances long-range dependency modeling. In the molecule-generation experiments, the authors highlight ring-closure errors as a challenging long-range dependency in SMILES. BARNN yields fewer ring-closure mistakes compared to baseline LSTMs, suggesting improved capacity to track tokens that appear far apart in the sequence.

4. BARNN-derived ensembles require fewer samples for stable statistical estimates and can revert to a single forward pass via MAP if uncertainty is not needed. The paper demonstrates that with about 30 ensemble members, the PDE solver’s predicted mean and variance converge. If uncertainty is not required, using the MAP estimate makes it as fast as a single forward pass, matching deterministic baselines on RMSE.

Overall, the claims are generally supported by well-chosen experiments that compare BARNN with multiple baselines. However, some details (e.g., how BARNN might scale to extremely large language models) are not deeply addressed, and real-world performance in large-scale domains is left for future work.

**Essential References Not Discussed:**

N/A

**Experimental Designs Or Analyses:**

- The PDE examples use well-known benchmark equations (Burgers, Kuramoto–Sivashinsky, Korteweg–de Vries) and evaluate multi-step rollouts (320 steps), which is a reasonable stress test.
- For molecule generation, a large ChEMBL-based dataset is used, covering real-world chemical diversity. Standard metrics in de novo drug design—validity, novelty, uniqueness, distribution matching—are computed.
- The comparison includes widely used baseline methods for uncertainty quantification (Dropout, Input Perturbation, etc.) and for molecule generation (standard RNNs).
- One limitation is that the PDE results rely on single-dimensional or low-dimensional PDE domains (1D PDE experiments). Similarly, the molecule generation uses a single tokenization approach (SMILES). Nonetheless, these design choices are typical entry points in the literature.

**Methods And Evaluation Criteria:**

The evaluation criteria appear sound for the chosen tasks. For additional confidence, the paper checks multiple seeds, compares ensemble sizes, and tests different PDEs. Overall, the methodology and metrics are consistent with accepted best practices in both scientific PDE modeling and generative modeling in chemistry.

**Other Comments Or Suggestions:**

It could be helpful to include an additional ablation that compares time-dependent dropout with a simpler “static” dropout factor, clarifying precisely how much performance gain is due to the time dimension vs. simply having an advanced prior.

**Other Strengths And Weaknesses:**

Strengths:
- The approach is elegant and general, allowing flexible application across domains.
- Empirical performance is strong, particularly regarding calibration and the ability to generate valid molecules.
- The derived prior (tVAMP) and time-dependent dropout mechanics feel both novel and easy to implement.

Weaknesses:
- The method’s computational overhead for extremely large models (like multi-billion-parameter LLMs) is not fully explored.
- Real-world PDEs often involve high-dimensional grids (2D, 3D); the current experiments may not fully reflect potential complexities in multi-dimensional domains.
- More thorough ablations could be done on alternative priors besides the new “tVAMP prior.”

**Questions For Authors:**

Why did you pick a tVAMP prior specifically, instead of simpler approximations (e.g., a fixed log-uniform prior for dropout)? Did you try them, and how did they perform?

**Relation To Broader Scientific Literature:**

This paper broadly builds on two themes: (1) Bayesian methods for deep networks (Variational Dropout, VAMP priors, etc.) and (2) autoregressive structures in PDEs, language modeling, or molecule generation.

**Theoretical Claims:**

1. The key theoretical contribution is a novel variational lower bound that explicitly models time-varying network weights and states, together with a new prior inspired by VAMP.
2. The authors provide a derivation that connects BARNN’s objective to the standard VAE framework.
3. The correctness of the proofs hinges on: factorization of the generative model over states and weights, the application of the local reparameterization trick for weight sampling, the derivation of the “best prior” in the sense of aggregated variational posteriors.

From a high-level view, the presented proofs appear logically consistent. However, a more detailed or step-by-step check of the proofs would help confirm there are no hidden assumptions (e.g., independence assumptions) that might be restrictive.

---

> ### Author Rebuttal · Authors · 2025-03-31
>
> We are delighted that the reviewer finds our approach *elegant and general, allowing flexible application across domains* and the tVAMP prior and time-dependent dropout mechanics *novel and easy to implement*. We appreciate that the reviewer considers that our *claims are generally supported by well-chosen experiments*. Finally, we thank the reviewer for the detailed comments and suggestions, which we address point by point below:
>
> **From a high-level view, the presented proofs appear logically consistent. However, a more detailed or step-by-step check of the proofs would help confirm there are no hidden assumptions that might be restrictive.**
> - We think the reviewer understood correctly the core of the proofs. For the temporal variational lower bound proof (Appendix A.1), given the specifics of the generative model distribution and the posterior distribution, no assumptions (e.g. independence or Markovian-memoryless properties) are made for deriving the lower bound. For the temporal VAMP prior proof (Appendix A.2), the approximation comes in eq. (19) while the general formulation of eq. (18) does not make any assumption.
>
> **Weaknesses:**
> **The method’s computational overhead for extremely large models (like multi-billion-parameter LLMs) is not fully explored**
> - We acknowledge the fact that we did not test, due to computational resources, the methodology on large-scale models. However, we want to highlight that our methodology can scale to large models due to the decoupling between posterior and main model weights. Indeed, the temporal dropout coefficients are obtained from a posterior model with shared weights, which contains order of magnitude lower number of parameters, compared to the main network which is left untouched in the parameter size.
>
> **Weaknesses:**
> **Real-world PDEs often involve high-dimensional grids (2D, 3D); the current experiments may not fully reflect potential complexities in multi-dimensional domains.**
> - We thank the reviewer for the suggestion, which we also find interesting to investigate. We believe our results will also scale to 2D and 3D experiments as we decouple the dropout rates from the main model weights. In our paper, we did not apply to 2D and 3D experiments for computational resources limitations (as GPU memory was limited).
>
> **Weaknesses:**
> **More thorough ablations could be done on alternative priors besides the new “tVAMP prior**
> - We thank the reviewer for this suggestion, which is shared with reviewer WqtE. To show how tVAMP excels over standard prior and fixed dropout rates, we conducted a test on a synthetic time-series dataset. This test was handmade to show how the prior affects the network modelling (i) for varying amplitudes and frequencies in the states, (ii) understand the effect of time adaptivitly of the prior. We also add a simpler static dropout model for reference in the hope of answering the question raised in Other Comments or Suggestions. The results report RMSE, NLL, and ECE statistics. *Static* method reports the RMSE obtained if the initial state is not propagated, while *MLP* is the base (non-bayesian) architecture (2 layer of with 64 with relu activation). BARNN uses the same MLP architecture, and it is ablate on different priors.
>
> Dataset:
> \begin{align}
> \begin{cases}
>     x_t &= x_{t-1} + \frac{3\pi}{100}, \quad x_0=0\\\\
>     y_t &=\frac{1}{5}\sum_{j=1}^5\sin(\alpha_i x_t +\beta_i), \quad \alpha_i\sim U[0.5, 1.5],\, \beta_i\sim U[0, 3\pi],\, t\in\{1, 2, \dots, 100\}
> \end{cases}
> \end{align}
>
> Table:
> | **Method**        | **Prior**      | **MSE** (↓)         | **NLL** (↓)           | **ECE** (↓)           |
> |-------------------|----------------|---------------------|-----------------------|-----------------------|
> | Static            | -              | 0.490 ± 0.000       | -                     | -                     |
> | MLP               | -              | 0.081 ± 0.011       | -                     | -                     |
> | Dropout (p=0.5)   | -              | 0.072 ± 0.004       | 0.593 ± 0.461         | 0.084 ± 0.010         |
> | Dropout (p=0.2)   | -              | 0.048 ± 0.004       | -0.075 ± 0.004        | 0.068 ± 0.009         |
> | BARNN             | log-uniform    | 0.045 ± 0.003       | -0.092 ± 0.064        | 0.050 ± 0.016         |
> | BARNN             | tVAMP          | **0.043 ± 0.001**   | **-0.166 ± 0.019**    | **0.049 ± 0.008**     |
>
> **Why did you pick a tVAMP prior specifically, instead of simpler approximations?....**
> - We thank the reveiwer for the interesting question. We picked the tVAMP prior because during preliminary experiments we found that using a standard log-uniform prior (commonly used for VD) did not give us good uncertainties, and we attribute it to the fact that the prior pushed to heavily spared network weights.  This is also confirmed by the time-series experiment results or the PDE (Table 1) results, where log-uniform and ARD priors obtain suboptimal results.

---

### Decision · Program_Chairs · 2025-05-01

**Decision:**

Accept (poster)

**Comment:**

This paper provides a framework to turn any autoregressive or recurrent neural network into a Bayesian version by incorporating a novel variational framing.  Specifically the authors define a temporal variational lower bound that they then sample from using a variant of variational dropout.   As the prior, the authors adapt the VAMP prior to a temporal setting (tVAMP), The authors show that their BARNN method outperforms  the non-Bayesian variant across a variety of experiments including PDE solvers, Molecule generation and some nice 1D uncertainty plots.

The scores were towards accept (4, 3, 3, 2) with an average leaning accept.  The reviewers seemed to find the paper well written and structured.  The method seemed sound, elegant and experiments are convincing.  In terms of weaknesses, the reviewers suggested that the method may not scale to very-large autoregressive models, e.g. billions of parameter models and one reviewer found that the method seemed like a somewhat straightforward application of variational dropout to temporal models.  The lowest review scored seemed to question novelty and seemed to want the authors to compare to some existing literature.  In general, however, the other reviewers found the contribution to be solid and the experiments convincing.  There were desires for some additional ablations, for example to validate the contribution of the tVAMP prior.

Overall this seems like a strong paper that addresses a significant problem of interest to the community (improving uncertainty of autoregressive models).  The method is quite general, not overly difficult, and empirically convincing.

Note there are some typos in the paper, e.g. multiple times "factories" was written instead of factorize.